# Don't Roll the Dice, Ask Twice: The Two-Query Distortion of Matching Problems and Beyond

**Georgios Amanatidis**
University of Essex
georgios.amanatidis@essex.ac.uk

**Georgios Birmpas**
Sapienza University of Rome
gebirbas@gmail.com

**Aris Filos-Ratsikas**
University of Edinburgh
Aris.Filos-Ratsikas@ed.ac.uk

**Alexandros A. Voudouris**
University of Essex
alexandros.voudouris@essex.ac.uk

## Abstract

In most social choice settings, the participating agents express their preferences over the different alternatives in the form of linear orderings. While this clearly simplifies preference elicitation, it inevitably leads to poor performance with respect to optimizing a cardinal objective, such as the social welfare, since the values of the agents remain virtually unknown. This loss in performance because of lack of information is measured by the notion of *distortion*. A recent array of works put forward the agenda of designing mechanisms that learn the values of the agents for a small number of alternatives via *queries*, and use this limited extra information to make better-informed decisions, thus improving distortion. Following this agenda, in this work we focus on a class of combinatorial problems that includes most well-known matching problems and several of their generalizations, such as One-Sided Matching, Two-Sided Matching, General Graph Matching, and $k$-Constrained Resource Allocation. We design *two-query* mechanisms that achieve the best-possible worst-case distortion in terms of social welfare, and outperform the best-possible expected distortion achieved by randomized ordinal mechanisms.

## 1 Introduction

The notion of *distortion* in social choice settings was defined to capture the loss in aggregate objectives due to the lack of precise information about the preferences of the participants [Procaccia and Rosenschein, 2006]. More concretely, the distortion was originally defined as a measure of the deterioration of the total happiness of the agents when access is given only to the (ordinal) preference rankings of the agents, rather than to the complete numerical (cardinal) information about their preferences. This research agenda has successfully been applied to a plethora of different settings, giving rise to a rich and vibrant line of work in major venues at the intersection of computer science and economics. For a comprehensive overview, see the survey of Anshelevich et al. [2021].

Out of all of these scenarios, some of the most fundamental are *matching* problems, in which agents are matched to items or other agents, aiming to maximize the *social welfare* of the matching (the total value of the agents). An example is the classic *One-Sided Matching* setting [Hylland and Zeckhauser, 1979], where the goal is to match $n$ items to $n$ agents based on the preferences of the agents over the items. For this setting, Filos-Ratsikas et al. [2014] showed that the best achievable distortion is

$\Theta(\sqrt{n})$. Importantly, this guarantee is only possible if one is allowed to use *randomization* and the values of the agents are *normalized*.[1]

Moving on from merely preference rankings, Amanatidis et al. [2021b] recently put forward the agenda of studying the tradeoffs between information and efficiency, when the employed mechanisms are equipped with the capability of learning the values of the agents via *queries*. The rationale is that asking the agents for more detailed information about only *a few* options is still cognitively not too burdensome, and could result in notable improvements on the distortion. This was indeed confirmed in that work for general social choice, and in a follow-up work for several matching problems [Amanatidis et al., 2021a]. Specifically, the latter work shows that it is possible to obtain distortion $O(n^{1/k})$ with $O(\log n)$ queries per agent for any constant integer $k$, and distortion $O(1)$ with $O(\log^2 n)$ queries per agent. Crucially, the mechanisms achieving these bounds do not use randomization nor demand the values to be normalized.

While these works make a significant first step, they leave some important questions unanswered. The mechanisms they propose require a logarithmic number of queries to achieve *any* significant improvement. Answering that many queries might still be cognitively too demanding for the agents, especially when there is a large number of possible options. The main high-level motivation of this research agenda, from our perspective, is that a small amount of information can be more valuable than randomization. But what does really constitute a "small amount"? Ideally, we would like to design mechanisms that make only a few queries per agent, independently of the size of the input parameters. Since with a single query, sub-linear distortion bounds are not possible [Amanatidis et al., 2021a,b], the first fundamental question that we would like to answer is the following:

*What is the best achievable distortion when we can only ask* two *queries per agent?*

## 1.1 Results and Technical Overview

We settle the aforementioned question for several matching problems, including *One-Sided Matching*, *Two-Sided Matching*, *General Graph Matching*, and other more general graph-theoretic problems. For all matching problems considered, we show that there is a deterministic mechanism that makes two queries per agent, runs in polynomial time, and achieves a distortion of $O(\sqrt{n})$. This upper bound is based on a novel mechanism, which we call MATCH-TWOQUERIES in the case of One-Sided Matching (see Mechanism 1). The mechanism asks two queries per agent and computes a maximum-weight matching based of the revealed values due to these queries. It starts by querying the agents at the first position of their preference rankings. For the second query, it computes a certain type of assignment $A$ of agents to items (or agents to agents in more general matching problems), to which we refer as a *sufficiently representative assignment*, and queries the agents about the items they are assigned to in $A$. The existence of such an assignment for all instances is far from trivial, and one of our main technical contributions is to show its existence and efficient computation for the wide range of problems we consider.

We also show that no deterministic mechanism for these settings that makes two queries per agent can achieve a distortion better than $\Omega(\sqrt{n})$. This lower bound follows by a more general construction yielding a lower bound of $\Omega(n^{1/\lambda})$ on the distortion of any mechanism that makes a constant number $\lambda$ of queries for any of these mechanisms. This mirrors the corresponding lower bounds of Amanatidis et al. [2021a] for One-Sided Matching.

While our results apply to general matching settings, their most impressive implications are for One-Sided Matching: We show that by using only *two* cardinal queries per agent, we can match the bound of $\Theta(\sqrt{n})$ for purely ordinal mechanisms, *without requiring randomization or any normalization*. MATCH-TWOQUERIES clearly also outperforms another mechanism of Amanatidis et al. [2021a], which uses two queries and achieves a distortion of $O(n^{2/3}\sqrt{\log n})$ assuming that the values of each agent sum up to $1$. In contrast, our mechanism works for unrestricted values, and achieves the best possible distortion of $O(\sqrt{n})$ based on conceptually much simpler ideas.

**Results for general social choice.** Given that our approach works for a wide variety of matching problems, one might be curious as to whether similar arguments could be used to show bounds for

---

[1]Note that if any of these assumptions is relaxed, it is impossible to achieve sub-linear distortion using only ordinal information.

the general social choice setting, where $n$ agents have preferences over $m$ alternatives, and the goal is to select an alternative with high social welfare; this was after all the original setting that Amanatidis et al. [2021b] studied in the introduction of the information-distortion tradeoff research agenda. In this setting, the situation is quite similar: the upper bounds follow by mechanisms that ask $O(\log m)$ queries, and nothing positive is known for smaller numbers of queries.

We show that a mechanism with structure similar to that of MATCH-TWOQUERIES can indeed achieve a distortion of $O(\sqrt{m})$ using only two queries, subject to being able to compute a sufficiently representative set of alternatives, which is analogous of the sufficiently representative assignment in matching problems. It turns out that this property is very closely connected to the notion of an *(approximately) stable committee* [Jiang et al., 2020, Cheng et al., 2020], and it follows that it exists when $m = \Omega(n)$, thus allowing us to obtain the desired bound of $O(\sqrt{m})$ when this is true. This case is quite natural, as it captures instances where a group of people need to decide over a large set of possible options (e.g., shortlisting candidates for a job, deciding the best paper for a conference, etc.). Interestingly, in contrast to the matching setting for which we show that sufficiently representative assignments can be found via a simple greedy algorithm, computing sufficiently representative sets of alternatives in general social choice requires rather involved techniques [Jiang et al., 2020, Cheng et al., 2020]. An obvious open question here is whether the $O(\sqrt{m})$ bound can also be achieved by asking only two queries when $m = o(n)$. This seems to be a more challenging task to prove; we discuss it further in Section 6.

We also show that the bound of $O(\sqrt{m})$ is the best possible, as part of a more general distortion lower bound of $\Omega(m^{1/\lambda})$ for mechanisms that make a constant number $\lambda$ of queries per agent; the latter result significantly improves the previously known lower bound of $\Omega(m^{1/2(\lambda+1)})$ [Amanatidis et al., 2021b].

**Roadmap.** For the sake of presentation, we fully demonstrate how our methodology works for the *One-Sided Matching* problem in Section 3. Before doing so, we start with some necessary notation and terminology in Section 2. In Section 4, we briefly discuss other graph-theoretic problems for which our methodology can be applied; all missing details are presented in the full version [Amanatidis et al., 2022]. Our results for general social choice are presented in Section 5; again, the detailed proofs are deferred to the full version. We conclude with some interesting open problems in Section 6.

## 1.2 Additional Related Work

The literature on the distortion of ordinal mechanisms in social choice is long and extensive, focusing primarily on settings with normalized utilities (e.g., [Boutilier et al., 2015, Caragiannis et al., 2017, Filos-Ratsikas et al., 2020]), or with metric preferences (e.g., [Anshelevich et al., 2018, Anshelevich and Postl, 2017, Gkatzelis et al., 2020]); see the survey of Anshelevich et al. [2021] for a detailed exposition. The distortion of mechanisms for One-Sided Matching and more general graph-theoretic problems has been studied in a series of works for a variety of preference models, but solely with ordinal information [Anshelevich and Zhu, 2018, Abramowitz and Anshelevich, 2018, Anshelevich and Zhu, 2017, Anshelevich and Sekar, 2016, Filos-Ratsikas et al., 2014, Caragiannis et al., 2016].

Besides the papers of Amanatidis et al. [2021b,a], the effect of limited cardinal information on the distortion has also been studied in other works [Abramowitz et al., 2019, Mandal et al., 2019, 2020, Benadè et al., 2021]. Mostly related to us is the paper of Ma et al. [2021] which considered the One-Sided Matching problem with a different type of cardinal queries, and showed qualitatively similar results to Amanatidis et al. [2021a] for Pareto optimality (rather than social welfare).

Our upper bound for the general social choice setting makes use of the results of Cheng et al. [2020] and Jiang et al. [2020] for (approximately) stable committees (see also Aziz et al. [2017]); a stable committee is very similar to a representative set of alternatives in our terminology. Cheng et al. [2020] showed that, while *exactly* stable committees do not always exist [Jiang et al., 2020], finding a random version of such committees, coined *stable lotteries*, is always possible and can be done in polynomial time. Later on, Jiang et al. [2020] showed that, via an intricate derandomization process, stable lotteries can yield approximately stable committees, where the approximation is a small multiplicative constant; for our purposes, this is sufficient. Interestingly, very recently, Ebadian et al. [2022] used stable lotteries to construct a purely ordinal randomized social choice mechanism that achieves the best possible distortion under unit-sum normalized values.

## 2 Preliminaries on One-Sided Matching, Mechanisms, and Distortion

In One-Sided Matching, there is a set $\mathcal{N}$ of $n$ *agents* and a set $\mathcal{A}$ of $n$ *items*. Each agent $i \in \mathcal{N}$ has a *value* $v_{i,j}$ for each item $j \in \mathcal{A}$; we refer to the matrix $\mathbf{v} = (v_{i,j})_{i \in \mathcal{N}, j \in \mathcal{A}}$ as the *valuation profile*. A *(one-sided) matching* $X : \mathcal{N} \to \mathcal{A}$ is a bijection from the set of agents to the set of items, i.e., each agent is *matched* to a different single item. Our goal is to choose a matching $X$ to maximize the *social welfare*, defined as the total value of the agents for the items they have been matched to according to $X$: $\text{SW}(X|\mathbf{v}) = \sum_{i \in \mathcal{N}} v_{i,X(i)}$. Usually $\mathbf{v}$ is clear from the context, so we then simplify our notation to $\text{SW}(X)$ for the social welfare of matching $X$.

As in most of the related literature, we assume that we do not have access to the valuation profile of the agents. Instead, we have access to the *ordinal preference* $\succ_i$ of each agent $i$, which is derived from the values of the agent for the items, such that $a \succ_i b$ if $v_{i,a} \geq v_{i,b}$; we refer to the vector $\succ_{\mathbf{v}} = (\succ_i)_{i \in \mathcal{N}}$ as the *ordinal profile* of the agents.

A *mechanism* $\mathcal{M}$ in our setting operates as follows: It takes as input the ordinal profile $\succ_{\mathbf{v}}$ of the agents. It then makes a number $\lambda \geq 1$ of *queries* per agent to learn part of the valuation profile. In particular, each agent is asked her value for at most $\lambda$ items. Given the answers to the queries, and also using the ordinal profile, $\mathcal{M}$ computes a feasible solution (here a matching) $\mathcal{M}(\succ_{\mathbf{v}})$.

In this paper we focus on mechanisms that make two queries per agent, i.e., $\lambda = 2$, and compute a solution of high social welfare. However, pinpointing an (approximately) optimal solution without having full access to the valuation profile of the agents can be quite challenging; the ordinal profile may be consistent with a huge number of different valuation profiles, even after the queries. Nevertheless, we aim to achieve the best asymptotic performance possible, as quantified by the notion of *distortion*.

**Definition 1.** The *distortion* of a mechanism $\mathcal{M}$ is the worst-case ratio (over the set $\mathcal{V}$ of all valuation profiles in instances with $n$ agents and $n$ items) between the optimal social welfare and the social welfare of the solution chosen by $\mathcal{M}$:

$$\text{dist}(\mathcal{M}) = \sup_{\mathbf{v} \in \mathcal{V}, |\mathcal{N}| = n, |\mathcal{A}| = n} \frac{\max_{X \in \mathcal{X}} \text{SW}(X|\mathbf{v})}{\text{SW}(\mathcal{M}(\succ_{\mathbf{v}})|\mathbf{v})},$$

where $\mathcal{X}$ is the set of all matchings between $\mathcal{N}$ and $\mathcal{A}$.

## 3 An Optimal Two-Query Mechanism

In this section, we present a mechanism for One-Sided Matching that makes two queries per agent and achieves a distortion of $O(\sqrt{n})$. Due to the lower bound of $\Omega(n^{1/\lambda})$ on the distortion of any mechanism that can make up to $\lambda$ queries per agent shown by Amanatidis et al. [2021a], our mechanism is asymptotically best possible when $\lambda = 2$.

Without any normalization assumptions about the valuation functions, it is easy to see that a mechanism cannot have *any* guarantee unless it queries every agent about her favorite item. However, there are no obvious criteria suggesting how to use the *second* query. Before we present the details of our mechanism, we define a particular type of assignment of agents to items that will be critical for deciding where to make the second queries.

**Definition 2.** A many-to-one assignment $A$ of agents to items (i.e., each agent is assigned to one item, but multiple agents may be assigned to the same item) is a *sufficiently representative assignment* if (a) For every item $j \in \mathcal{A}$, there are at most $\sqrt{n}$ agents assigned to $j$; (b) For any matching $X$, there are at most $\sqrt{n}$ agents that prefer the item they are matched to in $X$ to the item they are assigned to in $A$.

A natural question at this point is whether a sufficiently representative assignment exists for any instance, and if so, whether it can be efficiently computed. In Section 3.2, we present a simple polynomial-time algorithm for this task.

### 3.1 The Mechanism

Our mechanism MATCH-TWOQUERIES (Mechanism 1) first queries every agent about her favorite item. Next, it computes a sufficiently representative assignment $A$ (see Section 3.2) and queries each

agent about the item she is assigned to in $A$. Finally, it outputs a matching that maximizes the social welfare based *only* on the revealed values (all other values are set to 0). Although computational efficiency is not our primary focus here, if we use a polynomial-time algorithm for computing a maximum weight matching (e.g., the Hungarian method [Kuhn, 1956]), MATCH-TWOQUERIES runs in polynomial time as well.

---

**Mechanism 1** MATCH-TWOQUERIES($\mathcal{N}, \mathcal{A}, \succ_{\mathbf{v}}$)

---

1: Query each $i \in \mathcal{N}$ about her favorite item w.r.t. $\succ_i$
2: Compute a *sufficiently representative assignment* $A$
3: Query each agent about the item she is assigned to in $A$
4: Set all non-revealed values to 0
5: **return** a maximum-weight perfect matching $Y$

---

Of course, if we compare the mechanism's behaviour to an actual optimal matching $X$, we expect to see that we asked agents about the "wrong" items most of the time: for many agents the second query is about better items than what they are matched to in $X$, and for many agents it is about worse items. The desired bound of $O(\sqrt{n})$ on the distortion of MATCH-TWOQUERIES is established by balancing the loss due to each of these two cases.

**Theorem 1.** MATCH-TWOQUERIES *has distortion* $O(\sqrt{n})$.

*Proof.* Consider any instance with valuation profile $\mathbf{v}$. Let $Y$ be the matching computed by the MATCH-TWOQUERIES mechanism when given as input the ordinal profile $\succ_{\mathbf{v}}$, and let $X$ be an optimal matching. Let $\text{SW}_R(Y)$ be the *revealed* social welfare of $Y$, i.e., the total value of the agents for the items they are matched to in $Y$ and for which they *were queried* about. We will show that $\text{SW}(X) \leq (1 + 2\sqrt{n}) \cdot \text{SW}_R(Y)$, and then use the fact that $\text{SW}(Y) \geq \text{SW}_R(Y)$ to directly get the desired bound on the distortion.

We can write the optimal social welfare as

$$\text{SW}(X) = \text{SW}_R(X) + \text{SW}_C(X),$$

where $\text{SW}_R(X)$ is the revealed social welfare of $X$ that takes into consideration only the values revealed by the queries, whereas $\text{SW}_C(X)$ is the *concealed* social welfare of $X$ that takes into consideration only the values not revealed by any queries. Since $Y$ is the matching that maximizes the social welfare based only on the revealed values, we have that

$$\text{SW}_R(X) \leq \text{SW}_R(Y). \tag{1}$$

To bound the quantity $\text{SW}_C(X)$, let $S$ be the set of agents who are not queried about the items they are matched to in $X$. We partition $S$ into the following two subsets consisting of agents for whom the second query of the mechanism is used to ask about items that the agents consider *better* or *worse* than the items they are matched to in $X$. Recall that an agent $i$ is queried about the item $A(i)$ she is assigned to according to the sufficiently representative assignment $A$. So, $S$ is partitioned into

$$S^{\geq} = \left\{ i \in S : v_{i,A(i)} \geq v_{i,X(i)} \right\}, \quad \text{and} \quad S^{<} = \left\{ i \in S : v_{i,A(i)} < v_{i,X(i)} \right\}.$$

Given these sets, we can now write

$$\text{SW}_C(X) = \text{SW}_C^{\geq}(X) + \text{SW}_C^{<}(X),$$

where

$$\text{SW}_C^{\geq}(X) = \sum_{i \in S^{\geq}} v_{i,X(i)}$$

and

$$\text{SW}_C^{<}(X) = \sum_{i \in S^{<}} v_{i,X(i)}.$$

For every item $j$, let $S_j^{\geq} = \{i \in S^{\geq} : A(i) = j\}$ be the set of all agents in $S^{\geq}$ that are queried about $j$ by the mechanism using the second query. Thus, $S^{\geq} = \bigcup_{j \in \mathcal{A}} S_j^{\geq}$. Since $A$ is a sufficiently

representative assignment, $|S_j^{\geq}| \leq \sqrt{n}$ for every item $j$. Therefore,

$$
\begin{aligned}
\mathrm{SW}_C^{\geq}(X) = \sum_{j \in \mathcal{A}} \sum_{i \in S_j^{\geq}} v_{i,X(i)} &\leq \sum_{j \in \mathcal{A}} \sum_{i \in S_j^{\geq}} v_{i,j} \\
&\leq \sum_{j \in \mathcal{A}} |S_j^{\geq}| \cdot \max_{i \in S_j^{\geq}} v_{i,j} \leq \sqrt{n} \sum_{j \in \mathcal{A}} \max_{i \in S_j^{\geq}} v_{i,j} \\
&\leq \sqrt{n} \cdot \mathrm{SW}_R(Y). \tag{2}
\end{aligned}
$$

For the last inequality, recall that $A$ assigns every agent to a single item, and thus the sets $S_j^{\geq}$ are disjoint. In addition, the values of all the agents for the items they are matched to according to $A$ are revealed by the second query of the mechanism. Since we can match the agent in $S_j^{\geq}$ that has the maximum value for $j$ to $j$, and $Y$ maximizes the social welfare based on the revealed values, we obtain that $\mathrm{SW}_R(Y) \geq \sum_{j \in \mathcal{A}} \max_{i \in S_j^{\geq}} v_{i,j}$.

Next consider the quantity $\mathrm{SW}_C^{<}(X)$. By the fact that $A$ is a sufficiently representative assignment, it follows that $|S^{<}| \leq \sqrt{n}$; otherwise $X$ would constitute a matching for which there are strictly more than $\sqrt{n}$ agents that prefer the item they are matched to in $X$ to the item they are assigned to by $A$. Combined with the fact that all agents are queried at the first position of their ordinal preferences, we obtain

$$
\begin{aligned}
\mathrm{SW}_C^{<}(X) = \sum_{i \in S^{<}} v_{i,X(i)} &\leq \sum_{i \in S^{<}} \max_{j \in \mathcal{A}} v_{i,j} \\
&\leq |S^{<}| \cdot \max_{i \in S^{<}} \max_{j \in \mathcal{A}} v_{i,j} \\
&\leq \sqrt{n} \cdot \mathrm{SW}_R(Y). \tag{3}
\end{aligned}
$$

The bound follows directly by (1), (2) and (3). $\qquad \square$

## 3.2 Computing Sufficiently Representative Assignments

To establish the correctness of MATCH-TWOQUERIES, we need to ensure that a sufficiently representative assignment exists for any ordinal profile and that it can be computed efficiently. For this we present a simple polynomial time algorithm, which we call $\sqrt{n}$-SERIAL DICTATORSHIP (Mechanism 2). This algorithm creates $\sqrt{n}$ copies of each item and then runs a serial dictatorship algorithm, which first fixes an ordering of the agents and then assigns each agent to her most preferred available item according to her ordinal preference. It is easy to see that the running time of $\sqrt{n}$-SERIAL DICTATORSHIP is polynomial (in particular, it is $O(n^{1.5})$).

---

**Mechanism 2** $\sqrt{n}$-SERIAL DICTATORSHIP$(\mathcal{N}, \mathcal{A}, \succ_{\mathbf{v}})$

---

1: Let $\mathcal{B}$ be a multiset containing $\sqrt{n}$ copies of each $j \in \mathcal{A}$
2: **for** every agent $i \in \mathcal{N}$ **do**
3:      Let $\alpha_i$ be a most preferred item of agent $i$ in $\mathcal{B}$
4:      Remove $\alpha_i$ from $\mathcal{B}$
5: **end for**
6: **return** $A = (\alpha_i)_{i \in \mathcal{N}}$

---

**Theorem 2.** *For any instance, the output of $\sqrt{n}$-SERIAL DICTATORSHIP is a sufficiently representative assignment.*

*Proof.* Let $A$ be the output of the algorithm. During the execution of the algorithm, whenever every copy of an item has been assigned, we say that such an item is *exhausted*. Assume, towards a contradiction, that $A$ is not a sufficiently representative assignment. By construction, every item is assigned to at most $\sqrt{n}$ agents, so there must be a matching violating the second condition of Definition 2. That is, there is a subset of items $\mathcal{A}'$ and a subset of agents $\mathcal{N}'$, such that $|\mathcal{A}'| = |\mathcal{N}'| > \sqrt{n}$, and each agent $i \in \mathcal{N}'$ prefers to be assigned to a distinct item $\beta_i \in \mathcal{A}'$ (i.e., $\beta_i \neq \beta_j$ for $i \neq j$) instead of the item she is assigned to in $A$.

Consider any agent $i \in \mathcal{N}'$. The fact that this agent was not assigned to $\beta_i$ by the algorithm implies that when the agent was picked, item $\beta_i$ was exhausted. Since this is true for all agents in $\mathcal{N}'$, at the end of the algorithm all items of $\mathcal{A}'$ must be exhausted. However, an item is exhausted when all its $\sqrt{n}$ copies have been assigned and there are $n$ agents in total, so we can only have as many as $n/\sqrt{n} = \sqrt{n}$ exhausted items. This means that $|\mathcal{A}'| \leq \sqrt{n}$, a contradiction. $\qquad\square$

## 4 Further Combinatorial Optimization Problems

The approach of Section 3 can be extended to a much broader class of graph-theoretic problems. Informally, it works when the objective is to maximize an additive function over subgraphs of a given graph which contain all "small" matchings and have constant maximum degree. To make things more concrete, given a constant $k \in \mathbb{N}$ and a weighted graph $G$ on $n$ nodes, we say that a family $\mathcal{F}$ of subgraphs of $G$ is a *matching extending $k$-family* if:

- Graphs in $\mathcal{F}$ have maximum degree at most $k$;
- For any matching $M$ of $G$ of size at most $\lfloor n/3k \rfloor$, there is a graph in $\mathcal{F}$ containing $M$.

Clearly, the set of matchings of a graph is a matching extending 1-family, but it is not hard to see that different constraints are also captured, e.g., the set of subgraphs that are unions of disjoint paths and even cycles is a matching extending 2-family.

We are ready to introduce the general problem that we tackle here. As this is a special case of the class of problems captured by Ordinal-Max-on-Graphs (introduced by Amanatidis et al. [2021a]), we use a similar formulation and name.

***Ordinal-$k$-Max-on-Graphs***: Fix a constant $k \in \mathbb{N}$ and let $\mathcal{N}$ be a set of $n$ agents. Every agent $i \in \mathcal{N}$ has a (private) valuation function $v_i : \mathcal{N} \to \mathbb{R}_{\geq 0}$. We are given a graph $G = (\mathcal{N}, E)$, an ordinal profile $\succ_{\mathbf{v}} = (\succ_i)_{i \in \mathcal{N}}$ consistent with $\mathbf{v} = (v_i)_{i \in \mathcal{N}}$, and a concise description of a matching extending $k$-family $\mathcal{F}$. The goal is to find some $H \in \mathcal{F}$ that maximizes $\sum_{\{i,j\} \in E(H)} v_i(j)$.

Besides One-Sided Matching, a large number of problems that are relevant to computational social choice are captured by Ordinal-$k$-Max-on-Graphs. We give a few examples:

***General Graph $k$-Matching***: Given a graph $G$, find a $k$-matching of maximum value, i.e., $\mathcal{F}$ contains the subgraphs of $G$ of maximum degree at most $k$ and is a matching extending $k$-family. The problem becomes the ***General Graph Matching*** problem when $k = 1$; by also assuming that $G$ is bipartite, we have the celebrated ***Two-Sided Matching*** problem [Gale and Shapley, 1962, Roth and Sotomayor, 1992].

***$k$-Constrained Resource Allocation***: Given a bipartite graph $G = (\mathcal{N}_1 \cup \mathcal{N}_2, E)$, assign at most $k$ nodes of $\mathcal{N}_2$ to each node in $\mathcal{N}_1$ so that the total value of the corresponding edges is maximized. That is, $\mathcal{F}$ contains all 1-to-$k$ matchings of $G$ and is again a matching extending $k$-family. The problem for $k = 1$ becomes One-Sided Matching. Here $\mathcal{N}$ is partitioned into the set $\mathcal{N}_1$ of "actual agents" and the set $\mathcal{N}_2$ of "items", and $v_i(j)$ can be strictly positive only for $i \in \mathcal{N}_1, j \in \mathcal{N}_2$.

***Short Cycle Packing***: Given an integer $\ell$ and a weighted complete graph $G$, find a collection of node-disjoint cycles of length at most $\ell$ so that their total weight is maximized. Here, $\mathcal{F}$ is a matching extending 3-family (note that $k = 2$ for any $\ell$). It is worth mentioning that Short Cycle Packing is *not* a generalization of any of the matching problems above. It is also closely related to Clearing Kidney $\ell$-Exchanges [Abraham et al., 2007].

It is straightforward to extend the notion of distortion (Definition 1) for Ordinal-$k$-Max-on-Graphs by taking the supremum over all instances of a certain size $n$ and letting $\mathcal{X}$ be the set of feasible solutions of each instance.

As already discussed, for One-Sided Matching there is a lower bound of $\Omega(n^{1/\lambda})$ on the distortion of deterministic mechanisms that make up to $\lambda \geq 1$ queries per agent [Amanatidis et al., 2021a]. We can get the analogous result for *any* problem captured by Ordinal-$k$-Max-on-Graphs using a reduction that preserves distortion up to a constant factor. Further, by appropriately using the ideas of Section 3, we show that this lower bound is asymptotically tight for the case of two queries. For the statements of the theorems below, we assume that $k \in \mathbb{N}$ is a constant and that for every (general / bipartite) graph $G$, a matching extending $k$-family $\mathcal{F}(G)$ is specified.

**Theorem 3.** *No deterministic mechanism using at most $\lambda \geq 1$ queries per agent can achieve a distortion better than $\Omega(n^{1/\lambda})$ for Ordinal-k-Max-on-Graphs.*

For the positive result below, we rely on the existence of *a generalization* of sufficiently representative assignments. While the construction is very similar, the counting argument here is less intuitive compared to the case of One-Sided Matching.

**Theorem 4.** *There is a deterministic mechanism for Ordinal-k-Max-on-Graphs which uses at most two queries per agent and has distortion $O(\sqrt{n})$.*

Our mechanisms run in polynomial time whenever there is a polynomial-time algorithm for the full information version of the corresponding optimization problem. Luckily, all variants of matching problems we presented above can be solved efficiently by Edmond's algorithm [Edmonds, 1965] or its extensions [Marsh III, 1979].

**Corollary 5.** *There are deterministic polynomial-time mechanisms for General Graph Matching, Two-Sided Matching, General Graph k-Matching, and k-Constrained Resource Allocation which all use at most two queries per agent and have distortion $O(\sqrt{n})$.*

## 5 Towards Tight Bounds for General Social Choice

Here we consider the general social choice setting where a set $\mathcal{N}$ of $n$ *agents* have preferences over a set $\mathcal{A}$ of $m$ *alternatives*. As in the One-Sided Matching problem, there is a valuation profile $\mathbf{v} = (v_{i,j})_{i\in\mathcal{N},j\in\mathcal{A}}$ specifying the non-negative value that each agent $i$ has for every alternative $j$. The goal is to choose a single alternative $x \in \mathcal{A}$ to maximize the social welfare, that is, the total value of the agents for $x$: $\mathrm{SW}(x|\mathbf{v}) = \sum_{i\in\mathcal{N}} v_{i,x}$. Again, when $\mathbf{v}$ is clear from context, we will drop it from notation. Similarly to One-Sided Matching and the problems discussed in the previous section, $\mathbf{v}$ is unknown, and we are only given access to the ordinal profile $\succ_{\mathbf{v}}$ that is induced by $\mathbf{v}$. Social choice mechanisms must decide a single alternative based only on $\succ_{\mathbf{v}}$ and the values they can learn by making a small number of queries. The notion of distortion (Definition 1) can be extended for this setting as well, by taking the supremum over all instances with $n$ agents and $m$ alternatives, and letting $\mathcal{X}$ be the set $\mathcal{A}$ of alternatives.

For this general social choice setting, Amanatidis et al. [2021b] showed a lower bound of $\Omega\left(m^{1/(2(\lambda+1))}\right)$ on the distortion of mechanisms that make at most $\lambda \geq 1$ queries per agent. We improve this result by showing a lower bound of $\Omega(m^{1/\lambda})$ for any constant $\lambda$. The full proof of the following theorem can be found in the full version.

**Theorem 6.** *In the social choice setting, the distortion of any deterministic mechanism that makes at most a constant number $\lambda \geq 1$ of queries per agent is $\Omega(m^{1/\lambda})$.*

*Proof sketch.* Let $\mathcal{M}$ be an arbitrary mechanism that makes at most $\lambda \geq 1$ queries per agent. Consider the following instance with $n$ agents and $m = n$ alternatives. We assume that $m$ satisfies the condition $m \geq \frac{1}{2}\sum_{\ell=1}^{\lambda} m^{(\lambda-\ell+1)/\lambda} + 2$, and also that it is superconstant; otherwise the theorem holds trivially. We partition the set of alternatives $\mathcal{A}$ into $\lambda + 2$ sets $A_1, A_2, \dots A_{\lambda+1}, A_{\lambda+2}$, such that

- $|A_\ell| = \frac{1}{2}m^{(\lambda-\ell+1)/\lambda}$ for $\ell \in [\lambda]$;

- $|A_{\lambda+1}| = 2$;

- $|A_{\lambda+2}| = m - \frac{1}{2}\sum_{\ell=1}^{\lambda} m^{(\lambda-\ell+1)/\lambda} - 2$.

The ordinal profile has the following properties:

- For every $\ell \in [\lambda + 1]$, each alternative $j \in A_\ell$ is ranked at position $\ell$ by a set $T_{j,\ell}$ of $\frac{m}{|A_\ell|} = \Theta\left(m^{(\ell-1)/\lambda}\right)$ agents.

- For every $\ell \in [\lambda]$, every pair of agents that rank the same alternative in $A_\ell$ at position $\ell$, rank the same alternative in $A_{\ell+1}$ at position $\ell + 1$.

- For every agent, the alternatives that she does not rank in the first $\lambda + 1$ positions are ranked arbitrarily from position $\lambda + 2$ to $m$.

For every agent $i$, a query of $\mathcal{M}$ for alternative $j$ reveals a value of $m^{-\ell/\lambda}$ if $i$ ranks $j$ at position $\ell \in [\lambda + 1]$, and a value of $0$ if $i$ ranks $j$ at any other position.

Let $y$ be the alternative that $\mathcal{M}$ chooses as the winner for this instance. No matter the choice of $y$, we will define the cardinal profile so that it is consistent to the information revealed by the queries of $\mathcal{M}$, and the values of the agents for alternative $y$ are also consistent to the information that would have been revealed, irrespective of whether those values have actually been revealed. That is, any agent has a value of $m^{-\ell/\lambda}$ for $y$ if she ranks $y$ at position $\ell \in [\lambda + 1]$, and a value of $0$ if she ranks $y$ at any other position. Hence, the social welfare of $y$ is $\Theta\left(m^{(\ell-1)/\lambda}\right) \cdot m^{-\ell/\lambda} = \Theta(m^{-1/\lambda})$ if $y \in A_\ell$ for $\ell \in [\lambda + 1]$, or $0$ if $y \in A_{\lambda+2}$. Consequently, to show the desired bound of $\Omega(m^{1/\lambda})$ on the distortion of $\mathcal{M}$, it suffices to assume that $y \in A_\ell$ for some $\ell \in [\lambda + 1]$, and prove that the values of the agents that have not been revealed and do not correspond to alternative $y$ can always be defined such that there exists an alternative $x \neq y$ with social welfare $\Omega(1)$. The remaining details showing that such an $x$ always exists can be found in the full version. $\qquad\square$

Our approach for all the problems discussed in the previous sections can also be applied to the much more general social choice setting, *subject to* being able to compute a particular set of alternatives.

**Definition 3.** Let $c \geq 1$ be any constant. A subset of alternatives $B \subseteq \mathcal{A}$ with $|B| \leq c \cdot \sqrt{m}$ is a *sufficiently representative set* if, for every alternative $j \in \mathcal{A}$, at most $\sqrt{m}$ agents prefer $j$ over their favorite alternative in $B$.

We now present a mechanism that works *under the assumption* that sufficiently representative sets of alternatives can be (efficiently) computed; we discuss this assumption right after the statement of Theorem 7.

---
**Mechanism 3** SC-TWOQUERIES$(\mathcal{N}, \mathcal{A}, \succ_{\mathbf{v}})$

1: Query each agent about her favorite alternative
2: Compute a sufficiently representative set $B$
3: Query each agent for her favorite alternative in $B$
4: For every $j \in \mathcal{A}$, compute the revealed welfare $\mathrm{SW}_R(j)$
5: **return** $y \in \arg\max_{j \in \mathcal{A}} \mathrm{SW}_R(j)$

---

In particular, SC-TWOQUERIES (Mechanism 3) first queries each agent about her overall favorite alternative (the one ranked first). Then, given a sufficiently representative set of alternatives $B$, it queries each agent for her favorite alternative in $B$. Given the answers to these two queries per agent, the mechanism outputs an alternative that maximizes the *revealed* social welfare which is based only on the values learned from the queries.

**Theorem 7.** *The mechanism* SC-TWOQUERIES *has distortion* $O(\sqrt{m})$, *when restricted to the social choice instances for which a sufficiently representative set of alternatives exists.*

A sufficiently representative set of alternatives trivially exists when $m$ is much larger than $n$ (namely, when $m = \Omega(n^2)$). In contrast, when $m$ is much smaller than $n$, sufficiently representative sets of alternatives do not always exist.[2] Jiang et al. [2020] showed the following useful result:

**Theorem 8** ([Jiang et al., 2020]). *For any $\xi \in [n]$, there exists a set $S$ of alternatives with $|S| \leq 16 \cdot n/\xi$ such that for every $j \in A$, there are at most $\xi$ agents that prefer $j$ over their favorite alternative in $S$.*

A set $S$ as in the theorem above is called an *approximately stable committee* Cheng et al. [2020], Jiang et al. [2020]. Clearly, when $m = \Omega(n)$ and $\xi = \sqrt{n}$, an approximately stable committee is also

---
[2]For example, for any $k > \sqrt{m}$, consider an instance with $n = k \cdot m!$ agents, such that for each possible ordering of the $m$ alternatives there are exactly $k$ agents that have it as their preference. Then, for any subset $B$ of at most $\sqrt{m}$ alternatives and any alternative $j \in \mathcal{A} \setminus B$, there are at least $k > \sqrt{m}$ agents that prefer $j$ over any alternative in $B$.

a sufficiently representative set with $c = 16$. Therefore, combining Theorems 7 and 8, we obtain the following.

**Corollary 9.** *When $m = \Omega(n)$,* SC-TWOQUERIES *has distortion $O(\sqrt{m})$.*

## 6 Conclusion and Open Problems

In this paper, we showed that for a large class of problems, which includes One-Sided Matching and many other well-studied graph-theoretic problems, it is possible to achieve a distortion of $O(\sqrt{n})$ using a deterministic mechanism that makes at most two queries per agent, and that this is best possible asymptotically. Our whole methodology is based on computing assignments of agents to items or other agents that exhibit a very particular structure. In addition, in the social choice setting, when $m = \Omega(n)$, sets of alternatives with analogous properties can be computed, and our methodology yields a two-query mechanism with best possible distortion for this setting as well.

It is an interesting open problem to design a mechanism that makes two queries and achieves the best possible distortion of $O(\sqrt{m})$ when $m = o(n)$, or show that this is impossible. We suspect that to obtain a positive result one would need to come up with an adaptive mechanism, which decides where to ask each query based on the answers to all previous ones. Another question, about any of the problems we considered, is whether one can design mechanisms that make at most a constant $\lambda \geq 3$ queries per agent and their distortion matches the lower bound of $\Omega(n^{1/\lambda})$ (or, in the case of social choice, $\Omega(m^{1/\lambda})$).

## Acknowledgments and Disclosure of Funding

This work was supported by the ERC Advanced Grant 788893 AMDROMA "Algorithmic and Mechanism Design Research in Online Markets", the MIUR PRIN project ALGADIMAR "Algorithms, Games, and Digital Markets", and the NWO Veni project No. VI.Veni.192.153.

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
