*Appendix for:*

# Don't Roll the Dice, Ask Twice: The Two-Query Distortion of Matching Problems and Beyond

**Georgios Amanatidis**
University of Essex
georgios.amanatidis@essex.ac.uk

**Georgios Birmpas**
Sapienza University of Rome
gebirbas@gmail.com

**Aris Filos-Ratsikas**
University of Edinburgh
Aris.Filos-Ratsikas@ed.ac.uk

**Alexandros A. Voudouris**
University of Essex
alexandros.voudouris@essex.ac.uk

## A   Full Version of Section 4

As Section 4 of the main text is very condensed and only contains the necessary material to state Theorems 3 and 4, here we repeat everything in full detail.

We extend our results for One-Sided Matching to a much broader class of graph-theoretic problems. Informally, our approach works when the objective is to maximize an additive function over subgraphs of a given graph which contain all "small" matchings and have constant maximum degree. We make the space of feasible solutions more precise in the following definition.

**Definition 4.** Given a constant $k \in \mathbb{N}$ and a weighted graph $G$ on $n$ nodes, we say that a family $\mathcal{F}$ of subgraphs of $G$ is a *matching extending $k$-family* if:

- Graphs in $\mathcal{F}$ have maximum degree at most $k$;

- For any matching $M$ of $G$ of size at most $\lfloor n/3k \rfloor$, there is a graph in $\mathcal{F}$ containing $M$.

Clearly, the set of matchings of a graph (viewed as subgraphs rather than subsets of edges) is a matching extending 1-family, but it is not hard to see that Definition 4 captures other constraints, like subgraphs that are unions of disjoint paths and cycles (matching extending 3-family) or unions of disjoint cliques of size $k$ (matching extending $(k-1)$-family).

We are ready to introduce the general full information optimization problem that we tackle here; we then move on to its social choice analog. As this is a special case of the class of problems captured by Max-on-Graphs (introduced by Amanatidis et al. (2021)), we use a similar formulation and name. Note that, in the above definition, the family $\mathcal{F}$ is independent of the weights $w$. This is necessary as $w$ will be unknown in general.

***k-Max-on-Graphs***: Given a constant $k \in \mathbb{N}$, a weighted graph $G = (U, E, w)$, and a concise description of a matching extending $k$-family $\mathcal{F}$, find a solution $H^* \in \arg\max_{H \in \mathcal{F}} \sum_{e \in E(H)} w(e)$.

One-Sided Matching, as studied in Section 3, is the special case of $k$-Max-On-Graphs, where $G$ is the complete bipartite graph on the set of agents $\mathcal{N}$ and the set of items $\mathcal{A}$, the weight of an edge $\{i, j\}$ is the value $v_{i,j}$ of agent $i$ for item $j$, and $\mathcal{F}$ contains all the 1-factors of $G$. Note that the weights of the graph in this case are defined in terms of the valuation functions of the agents. Moreover, recall that in our setting only the ordinal preferences of the agents are given and their cardinal values can be accessed only via queries; so, we do not know the weights that have not been revealed by a query. This is the case for all the problems we are interested in, and is captured by the next definition. To

36th Conference on Neural Information Processing Systems (NeurIPS 2022).

avoid unnecessary notation, items are modeled as dummy agents with all their cardinal values equal to 0. In addition, we write $w(H) := \sum_{e \in E(H)} w(e)$.

***Ordinal-k-Max-on-Graphs***: Fix a constant $k \in \mathbb{N}$ and let $\mathcal{N}$ be a set of $n$ agents. A weighted graph $G = (\mathbb{N}, E, w)$ is given *without* its weights. Every agent $i \in \mathcal{N}$ has a (private) valuation function $v_i : \mathbb{N} \to \mathbb{R}_{\geq 0}$, such that, for every $e = \{i, j\} \in E$,

$$w(e) = v_i(j) + v_j(i).$$

We are also given an *ordinal profile* $\succ_{\mathbf{v}} = (\succ_i)_{i \in \mathcal{N}}$ that is consistent to $\mathbf{v} = (v_i)_{i \in \mathcal{N}}$, and a concise description of a matching extending $k$-family $\mathcal{F}$. The goal is to find $H^* \in \arg\max_{H \in \mathcal{F}} w(H)$.

Besides One-Sided Matching, a large number of problems that are relevant to computational social choice are captured by Ordinal-$k$-Max-on-Graphs. We give a few examples:

***General Graph Matching***: Given a weighted graph $G = (U, E, w)$, find a matching of maximum weight, i.e., $\mathcal{F}$ contains the matchings of $G$ and clearly is a matching extending 1-family. In the social choice analog of the problem, $U = \mathcal{N}$.

***Two-Sided Matching***: This is a special case of General Graph Matching in which $G = (U_1 \cup U_2, E, w)$ is a bipartite graph. It is an extensively studied problem in economics and computational social choice (Gale and Shapley, 1962; Roth and Sotomayor, 1992).

***k-Clique Packing***: Given a weighted complete graph $G = (U, E, w)$, where $|U| = n$ is a multiple of $k \geq 2$, the goal is to partition $U$ into $|U|/k$ clusters of size $k$ to maximize the total weight of the edges in the clusters. That is, $\mathcal{F}$ contains all spanning subgraphs of $G$ that are the union of cliques of size $k$. As claimed above, $\mathcal{F}$ is a matching extending $(k-1)$-family: clearly every graph in $\mathcal{F}$ has maximum degree $k - 1$, and any matching of size $\lfloor n/(3(k-1)) \rfloor$ (which is less than $n/k$) can be extended to a graph in $\mathcal{F}$ by arbitrarily grouping each pair of matched nodes with $k - 2$ unmatched nodes, and then arbitrarily grouping the remaining nodes $k$ at a time.

This problem generalizes General Graph Matching (for which $k = 2$) and often is referred to as Max $n/k$-Sum Clustering in the literature; see (Anshelevich and Sekar, 2016). In its social choice analog, $U = \mathcal{N}$.

***General Graph k-Matching***: Given a weighted graph $G = (U, E, w)$, find a $k$-matching of maximum weight, i.e., $\mathcal{F}$ contains all the subgraphs of $G$ where each node has degree at most $k$. As $\mathcal{F}$ already contains all matchings of $G$ of any size, it is straightforward that it is a matching extending $k$-family. In the social choice analog of the problem, $U = \mathcal{N}$.

***k-Constrained Resource Allocation***: Given a bipartite weighted graph $G = (U_1 \cup U_2, E, w)$, the goal is to assign at most $k$ nodes of $U_2$ to each node in $U_1$ so that the total weight of the corresponding edges is maximized. That is, $\mathcal{F}$ contains the subgraphs of $G$ where each node in $U_1$ has degree at most $k$ and each node in $U_2$ has degree at most 1. Again, $\mathcal{F}$ already contains all matchings of $G$ of any size, so it is a matching extending $k$-family.

This problem generalizes One-Sided Matching. In its social choice analog, $\mathcal{N} = U_1 \cup U_2$ is partitioned into the "actual agents" $\mathcal{N}_1 = U_1$ and the "items" $\mathcal{N}_2 = U_2$, and $v_i(j)$ can be strictly positive only for $i \in \mathcal{N}_1, j \in \mathcal{N}_2$.

***Short Cycle Packing***: Given an integer $\ell$ and a weighted complete graph $G = (U, E, w)$, the goal is to find a collection of node-disjoint cycles of length at most $\ell$ so that their total weight is maximized. Here, $\mathcal{F}$ contains any such collection of short cycles. Arguing as in $k$-Clique Packing, it is straightforward to see that $\mathcal{F}$ is a matching extending $(\ell - 1)$-family (although it is not hard to show that it is actually a matching extending 3-family). The social choice analog of the problem has $U = \mathcal{N}$, and is closely related to Clearing Kidney $\ell$-Exchanges (Abraham et al., 2007).

As already discussed in the Introduction, for One-Sided Matching, Amanatidis et al. (2021) showed a lower bound of $\Omega(n^{1/\lambda})$ on the distortion of all deterministic mechanisms that can make up to $\lambda \geq 1$ queries per agent. Using this, we can get the analogous result for all the aforementioned problems. Although for some of them, like Two-Sided Matching and General Graph Matching, the lower bound is immediate, here we show it for *any* problem captured by Ordinal-$k$-Max-on-Graphs. For the statement of the theorem, $k \in \mathbb{N}$ is a constant, and we assume that for every graph $G$ a matching extending $k$-family $\mathcal{F}(G)$ is specified.

**Theorem 3.** *No deterministic mechanism using at most $\lambda \geq 1$ queries per agent can achieve a distortion better than $\Omega(n^{1/\lambda})$ for Ordinal-k-Max-on-Graphs with feasible solutions given by $\mathcal{F}(\cdot)$.*

*Proof.* We are going to show that if we had a deterministic mechanism $\mathcal{M}$ for Ordinal-$k$-Max-on-Graphs that makes at most $\lambda \geq 1$ queries per agent and achieves distortion $o(n^{1/\lambda})$, then we could design a deterministic mechanism for One-Sided Matching that also makes at most $\lambda$ queries per agent and has distortion $o(n^{1/\lambda})$. As the latter is impossible (Amanatidis et al., 2021), that would imply that the lower bound applies to Ordinal-$k$-Max-on-Graphs as well.

Let $(\mathcal{N}, \mathcal{A}, \succ_{\mathbf{v}})$ be an arbitrary instance of One-Sided Matching with $|\mathcal{N}| = |\mathcal{A}| = n$ and underlying weights defined by $\mathbf{v} = (v_{i,j})_{i \in \mathcal{N}, j \in \mathcal{A}}$. We essentially use the same instance for Ordinal-$k$-Max-on-Graphs: A complete bipartite weighted graph $G = (U_1 \cup U_2, E, w)$ with $U_1 = \mathcal{N}$, $U_2 = \mathcal{A}$, and valuation functions defined as $u_i(j) = v_{i,j}$ and $u_j(i) = 0$ for every $i \in \mathcal{N}, j \in \mathcal{A}$; the induced ordinal profile is also well-defined. Clearly, the matchings in the two instances are exactly the same and have the same weight (although they may not be feasible with respect to $\mathcal{F}(G)$). However, the feasible solutions for Ordinal-$k$-Max-on-Graphs include subgraphs where the nodes may have degree up to $k$ instead of $1$. We need to relate the weight of an (approximately) optimal solution for the Ordinal-$k$-Max-on-Graphs instance to the value of an optimal matching for the One-Sided Matching instance.

Let $M$ be a maximum weight matching in $G$ (and thus a maximum-value matching for the original One-Sided Matching instance) and $H \in \mathcal{F}(G)$ be an optimal solution. Consider the submatching $\hat{M}$ of $M$ that uses the $\lfloor 2n/3k \rfloor$ heaviest edges of $M$. Using the fact that $\lfloor x \rfloor \geq x/2$ for $x \geq 1$, we get

$$w(\hat{M}) \geq \frac{\lfloor 2n/3k \rfloor}{n} w(M) \geq \frac{1}{3k} w(M). \tag{4}$$

Since $\mathcal{F}(G)$ is a matching extending $k$-family and $\hat{M}$ is sufficiently small (since $|V(G)| = 2n$), there is some $\hat{H} \in \mathcal{F}(G)$ such that $\hat{M}$ is a subgraph of $\hat{H}$. As $H$ is a maximum-weight element of $\mathcal{F}(G)$, we directly get $w(H) \geq w(\hat{H}) \geq w(\hat{M})$ and, combining with (4), we have

$$w(H) \geq \frac{1}{3k} w(M). \tag{5}$$

Now, if $H' \in \mathcal{F}(G)$ is an $\alpha$-approximate solution to the same Ordinal-$k$-Max-on-Graphs instance, then (5) implies

$$w(H') \geq \frac{1}{\alpha} w(H) \geq \frac{1}{3\alpha k} w(M). \tag{6}$$

We can construct a matching from $H'$ using only ordinal information. In particular, for each $i \in U_1$, among the edges in $H'$ that are incident to $i$, we keep the best one with respect to $\succ_i$. Of course, the resulting graph $H''$ may not be a matching, as each node in $U_2$ may still have degree up to $k$. However, note that this process also keeps at least a $1/k$ fraction of the weight incident to each $i \in U_1$, and thus of the total weight. So, (6) implies

$$w(H'') \geq \frac{1}{k} w(H') \geq \frac{1}{3\alpha k^2} w(M).$$

We repeat the process for the remaining nodes: for each $j \in U_2$, we keep the best of its edges in $H''$ with respect to $\succ_j$. Now the resulting graph $M'$ *is* a matching and has at least a $1/k$ fraction of the total weight of $H''$ and, thus,

$$w(M') \geq \frac{1}{k} w(H'') \geq \frac{1}{3\alpha k^3} w(M).$$

If needed, we can extend $M'$ to a perfect matching $M''$ by arbitrarily matching the unmatched nodes of $U_1$ and $U_2$, and consider its analog in the original instance. Clearly, $w(M'') \geq w(M')$, and thus $M''$ is a $(3\alpha k^3)$-approximate solution for the original One-Sided Matching instance. Therefore, if there existed a mechanism $\mathcal{M}$ with distortion $\alpha = o(n^{1/\lambda})$ for Ordinal-$k$-Max-on-Graphs with feasible solutions given by $\mathcal{F}$, we could use it for the above instance to get $H'$ and then $M''$, which would have weight within a factor of $o(3k^3 n^{1/\lambda})$ from a maximum weight matching. Since $k$ is a constant, this would imply a distortion of $o(n^{1/\lambda})$ for One-Sided Matching, a contradiction. $\square$

We are particularly interested in the case of $\lambda = 2$. In the next two sections we are going to present a mechanism for this case, which is asymptotically optimal, namely it achieves distortion $O(\sqrt{n})$, matching the lower bound we just derived.

## A.1 Sufficiently Representative Assignments

We now revisit the notion of a *sufficiently representative assignment*. We will appropriately adjust it to refer to a single set of agents (which, for the case of One-Sided Marching, includes both the actual agents and the items), and also incorporates the parameter $k$ from the definition of Ordinal-$k$-Max-on-Graphs.

**Definition 5.** Given $\mathcal{N}_1, \mathcal{N}_2 \subseteq \mathcal{N}$ and $k \in \mathbb{N}$, a many-to-one assignment $A$ of agents in $\mathcal{N}_1$ to agents in $\mathcal{N}_2$ is an $(\mathcal{N}_1, \mathcal{N}_2, k)$-*sufficiently representative assignment* if:

- For every agent $j \in \mathcal{N}_2$, there are at most $\sqrt{n}$ agents from $\mathcal{N}_1$ assigned to $j$;

- For any bipartite graph $H$ with node set $\mathcal{N}_1 \cup \mathcal{N}_2$ and maximum degree $k$, there are at most $k\sqrt{n}$ agents in $\mathcal{N}_1$ that prefer some of their neighbors in $H$ to the agent they are assigned to according to $A$.

For One-Sided Matching and $k$-Constrained Resource Allocation, $\mathcal{N}_1$ is the set of actual agents and $\mathcal{N}_2$ is the set of items. In contrast, for all other problems considered here we have $\mathcal{N}_1 = \mathcal{N}_2 = \mathcal{N}$.

Like we did in Section 3.2, we need to show that an $(\mathcal{N}_1, \mathcal{N}_2, k)$-sufficiently representative assignment exists for any instance of Ordinal-$k$-Max-on-Graphs and any $\mathcal{N}_1, \mathcal{N}_2 \subseteq \mathcal{N}$. We rely on the same high-level idea for the construction: $\sqrt{n}$ copies of each agent in $\mathcal{N}_2$ are created and then a SERIAL DICTATORSHIP algorithm is run with respect to the agents in $\mathcal{N}_1$. The running time of $\sqrt{n}$-SERIAL DICTATORSHIP remains $O(n^{1.5})$.

---

**Mechanism 1** $\sqrt{n}$-SERIAL DICTATORSHIP$(\mathcal{N}_1, \mathcal{N}_2, \succ_{\mathbf{v}})$

---

1: Let $\mathcal{B}$ be a multiset with $\sqrt{n}$ copies of each $j \in \mathcal{N}_2$
2: **for** every agent $i \in \mathcal{N}_1$ **do**
3:     Let $\alpha_i$ be $i$'s most preferred agent in $\mathcal{B}$ w.r.t. $\succ_i$
4:     Remove $\alpha_i$ from $\mathcal{B}$
5: **end for**
6: **return** $A = (\alpha_i)_{i \in \mathcal{N}_1}$

---

The following theorem is the analog of Theorem 2. While the proof is very similar, the counting argument here is somewhat less intuitive compared to the case of One-Sided Matching due to the differences between Definitions 2 and 5.

**Theorem 10.** *The assignment computed by the* $\sqrt{n}$-SERIAL DICTATORSHIP *algorithm is an* $(\mathcal{N}_1, \mathcal{N}_2, k)$-*sufficiently representative assignment.*

*Proof.* Let $A$ be the assignment produced by the algorithm. During the execution of the algorithm, whenever all the copies of an agent $j \in \mathcal{N}_2$ have been matched, we will say that $j$ is *exhausted*. Assume towards a contradiction that $A$ is not an $(\mathcal{N}_1, \mathcal{N}_2, k)$-sufficiently representative assignment. By construction, the first condition of Definition 5 is obviously satisfied. So, there must be a graph $H$, as described in the second condition of Definition 5, with respect to which there exists a subset $\mathcal{S}_1 \subseteq \mathcal{N}_1$ with $|\mathcal{S}_1| > k\sqrt{n}$ such that every $i \in \mathcal{S}_1$ prefers her best neighbor in $H$, say $\beta_i$, to agent $\alpha_i$ she has been assigned to in $A$. Let $\mathcal{S}_2 \subseteq \mathcal{N}_2$ be the set that contains all these $\beta_i$s. Because $H$ has maximum degree at most $k$ we have $|\mathcal{S}_2| \geq |\mathcal{S}_1|/k > \sqrt{n}$.

Consider any agent $i \in \mathcal{S}_1$. The fact that this agent was not assigned to $\beta_i$ by $\sqrt{n}$-SERIAL DICTATORSHIP implies that when it was $i$'s turn to pick, agent $\beta_i$ was exhausted. Therefore, at the end of the algorithm, all agents of $\mathcal{S}_2$ must be exhausted. Since an agent in $\mathcal{N}_2$ is exhausted when all its $\sqrt{n}$ copies have been assigned and there are at most $n$ agents in $\mathcal{N}_1$, we can only have as many as $\frac{n}{\sqrt{n}} = \sqrt{n}$ exhausted agents. The latter means that $|\mathcal{S}_2| \leq \sqrt{n}$, a contradiction. $\qquad\square$

## A.2 The General Mechanism

We are now ready to show that it is possible to achieve distortion $O(\sqrt{n})$ for *any* problem that can be modeled as a special case of Ordinal-$k$-Max-on-Graphs. Our mechanism generalizes the main idea of The Mechanism of querying each agent about for her overall favorite alternative, as well

as the alternative suggested by an appropriate sufficiently representative assignment. For the latter, we need to specify $\mathcal{N}_1$ and $\mathcal{N}_2$: These are typically both equal to the whole $\mathcal{N}$, unless the problem distinguishes between actual agents and items, in which case these two groups are captured by $\mathcal{N}_1$ and $\mathcal{N}_2$, respectively. In any case, all the edges in a feasible solution have at least one endpoint in each of $\mathcal{N}_1$ and $\mathcal{N}_2$.

---

**Mechanism 2** GENERAL-TWOQUERIES$(G, \mathcal{F}, \mathcal{N}_1, \mathcal{N}_2, \succ_{\mathbf{v}})$

---

1: Query each agent $i \in \mathcal{N}_1$ for her favorite alternative in $\mathcal{N}_2$ w.r.t. $\succ_i$
2: Compute $A = \sqrt{n}$-SERIAL DICTATORSHIP$(\mathcal{N}_1, \mathcal{N}_2, \succ_{\mathbf{v}})$
3: Query each agent $i \in \mathcal{N}_1$ about the agent $\alpha_i \in \mathcal{N}_2$ she is assigned to in $A$
4: Set all non-revealed values to $0$
5: **return** a maximum-weight member of $\mathcal{F}$

---

Note that the final step of the algorithm involves computing a solution that is optimal according to the revealed values. There are computational issues to consider here, however, we first tackle the question of whether it is even possible to match the lower bounds of Theorem 3 for $\lambda = 2$, despite the lack of information. We briefly discuss how to transform GENERAL-TWOQUERIES into a polynomial-time mechanism after the proof of Theorem 4 below. Again, for the statement of the theorem we assume that $k \in \mathbb{N}$ is a constant and that, for every $G$, a matching extending $k$-family $\mathcal{F}(G)$ is specified.

**Theorem 4.** *For Ordinal-k-Max-on-Graphs with feasible solutions given by $\mathcal{F}(\cdot)$, GENERAL-TWOQUERIES has distortion $O(\sqrt{n})$.*

*Proof.* Consider any instance with valuation profile $\mathbf{v}$ and relevant sets of agents $\mathcal{N}_1$ and $\mathcal{N}_2$. Let $Y$ be the solution computed by the GENERAL-TWOQUERIES mechanism when given as input $(G, \mathcal{F}, \mathcal{N}_1, \mathcal{N}_2, \succ_{\mathbf{v}})$, and let $X$ denote an optimal solution. Let $w_R(Y)$ be the *revealed* weight of $Y$ as seen by the mechanism, that is, the weight of $Y$ taking into account only the values that have been revealed by the queries. We will show that $w(X) \leq (1 + 10k^2\sqrt{n}) \cdot w_R(Y)$, and the bound on the distortion will then follow by the obvious fact that $w(Y) \geq w_R(Y)$.

We can write the optimal weight as

$$w(X) = w_R(X) + w_C(X), \tag{7}$$

where $w_R(X)$ is the revealed weight of $X$ that takes into account only the values that have been revealed by the queries, whereas $w_C(X)$ is the *concealed* weight of $X$ that takes into account only the values that have not been revealed by the queries of the mechanism. Since $Y$ is the solution that maximizes the social welfare based only on the revealed values, we have that

$$w_R(X) \leq w_R(Y). \tag{8}$$

Thus, it suffices to bound $w_C(X)$.

Let $S$ be the set of agents in $\mathcal{N}_1$ who are not queried about all their neighbors in $X$. We partition $S$ into two subsets $S^{\geq}$ and $S^{<}$ consisting of agents for whom the second query of the mechanism is used to ask about someone they consider *better* or *worse* than their best neighbor in $X$, respectively. For an agent $i \in \mathcal{N}_1$, let $\chi_i$ be $i$'s favorite neighbor in $X$ and recall that $i$ is queried about agent $\alpha_i \in \mathcal{N}_2$ to whom she is assigned according to the $(\mathcal{N}_1, \mathcal{N}_2, k)$-sufficiently representative assignment $A$. So,

$$S^{\geq} = \left\{ i \in S : v_{i,\alpha_i} \geq v_{i,\chi_i} \right\},$$
$$S^{<} = \left\{ i \in S : v_{i,\alpha_i} < v_{i,\chi_i} \right\}.$$

Let $N_X(i)$ be the set of agents who are neighbors of $i$ in $X$ *and* for which $i$ was not queried about. We now define

$$w_C^{\geq}(X) = \sum_{i \in S^{\geq}} \sum_{j \in N_X(i)} v_{i,j}$$
$$w_C^{<}(X) = \sum_{i \in S^{<}} \sum_{j \in N_X(i)} v_{i,j}$$

Clearly, $w_C(X) = w_C^{\geq}(X) + w_C^{<}(X)$.

For every agent $j \in \mathcal{N}_2$, let $S_j^{\geq} = \{i \in S^{\geq} : \alpha_i = j\}$ be the set of all agents in $S^{\geq}$ that are queried about $j$ by the mechanism using the second query. So, $S^{\geq} = \bigcup_{j \in \mathcal{N}_2} S_j^{\geq}$. Since $A$ is an $(\mathcal{N}_1, \mathcal{N}_2, k)$-sufficiently representative assignment, the first condition of Definition 5 implies that $|S_j^{\geq}| \leq \sqrt{n}$ for every $j \in \mathcal{N}_2$. Therefore,

$$
\begin{aligned}
w_C^{\geq}(X) &= \sum_{j \in \mathcal{N}_2} \sum_{i \in S_j^{\geq}} \sum_{\ell \in N_X(i)} v_{i,\ell} \\
&\leq \sum_{j \in \mathcal{N}_2} \sum_{i \in S_j^{\geq}} \sum_{\ell \in N_X(i)} v_{i,j} \\
&\leq \sum_{j \in \mathcal{N}_2} \sum_{i \in S_j^{\geq}} k \cdot v_{i,j} \\
&\leq k \sum_{j \in \mathcal{N}_2} |S_j^{\geq}| \max_{i \in S_j^{\geq}} v_{i,j} \\
&\leq k\sqrt{n} \sum_{j \in \mathcal{N}_2} \max_{i \in S_j^{\geq}} v_{i,j}
\end{aligned}
\tag{9}
$$

where the first inequality holds by the definition of the sets $S_j^{\geq}$, for every $j \in \mathcal{N}_2$. To complete our bound on $w_C^{\geq}(X)$, we need the following claim.

**Claim 1.** *For all the problems of interest,*

$$
\sum_{j \in \mathcal{N}_2} \max_{i \in S_j^{\geq}} v_{i,j} \leq 9k \cdot w_R(Y).
$$

*Proof.* For each $j \in \mathcal{N}_2$, let $i_j \in \arg\max_{i \in S_j^{\geq}} v_{i,j}$. Consider the subgraph $H$ of the input graph $G$ with edge set $E(H) = \{\{i_j, j\} \mid j \in \mathcal{N}_2\}$, i.e., $H$ contains exactly the edges that define the sum of interest. In particular, we have

$$
\sum_{j \in \mathcal{N}_2} \max_{i \in S_j^{\geq}} v_{i,j} = \sum_{j \in \mathcal{N}_2} v_{i_j,j} = w_R(H).
$$

We now claim that each node in $H$ has degree at most 2. To see this, consider an agent $\ell \in \mathcal{N}$. There is at most one $j \in \mathcal{N}_2$ such that $\ell \in S_j^{\geq}$ (since these sets are disjoint), and thus we may have $\ell = i_j$ for at most one $j \in \mathcal{N}_2$, resulting in the edge $\{j, \ell\}$ in $H$. Additionally, $\ell$ may itself be in $\mathcal{N}_2$, resulting in a second edge $\{\ell, i_\ell\}$ in $H$. Other than these two, there can be no other edges of $H$ adjacent to $\ell$.

Since $H$ has maximum degree at most 2, it must contain a matching $M$ of comparable weight. Specifically, $H$ must be the union of node-disjoint paths and cycles. We construct a (possibly empty) matching $M_1$ on $H$ by arbitrarily picking one edge from each odd cycle and one edge from the beginning of each odd path. If we remove $M_1$ from $H$, then the remaining graph is the union of node-disjoint even paths and even cycles, and thus can be decomposed into two disjoint matchings $M_2, M_3$ in a straightforward way. Since $M_1, M_2,$ and $M_3$ cover all the edges of $H$, the best of them, say $M$, must have weight at least $w_R(H)/3$, i.e,

$$
w_R(M|\mathbf{v}) = \frac{1}{3} w_R(H).
$$

Now we can work with $M$ like in the proof of Theorem 3. Consider the submatching $\hat{M}$ of $M$ containing the $\lfloor 2n/3k \rfloor$ heaviest edges of $M$ to get

$$
w_R(\hat{M}) \geq \frac{1}{3k} w_R(M) \geq \frac{1}{9k} w_R(H).
\tag{10}
$$

Since $\mathcal{F}$ is a matching extending $k$-family and $\hat{M}$ is sufficiently small, there is some $\hat{Y} \in \mathcal{F}(G)$ such that $\hat{M}$ is a subgraph of $\hat{Y}$. As $Y$ is a maximum-weight element of $\mathcal{F}(G)$ with respect to the revealed weights, we directly get $w_R(Y) \geq w_R(\hat{Y}) \geq w_R(\hat{M})$ and, combining with (10), we have

$$w_R(Y) \geq \frac{1}{9k} w(H),$$

as desired. □

By combining (9) with Claim 1, we get

$$w_C^{\geq}(X) \leq 9k^2 \sqrt{n} \cdot w_R(Y). \tag{11}$$

We next consider the quantity $w_C^{<}(X)$. By the fact that $A$ is an $(\mathcal{N}_1, \mathcal{N}_2, k)$-sufficiently representative assignment, it follows that $|S^{<}| \leq k\sqrt{n}$; otherwise $X$ would be a graph that violates the second condition of Definition 5. Combined with the fact that all agents in $\mathcal{N}_1$ are queried about their favorite alternative, we can obtain the following upper bound on $w_C^{<}(X)$. Recall that for $i \in \mathcal{N}_1$, we have $N_X(i) \subseteq \mathcal{N}_2$ and $|N_X(i)| \leq k$.

$$\begin{aligned}
w_C^{<}(X) &= \sum_{i \in S^{<}} \sum_{j \in N_X(i)} v_{i,j} \\
&\leq \sum_{i \in S^{<}} k \cdot \max_{j \in \mathcal{N}} v_{i,j} \\
&\leq k \, |S^{<}| \max_{i \in S^{<}} \max_{j \in \mathcal{N}} v_{i,j} \\
&\leq k^2 \sqrt{n} \cdot w_R(Y).
\end{aligned} \tag{12}$$

The bound now follows by (7), (8), (11), (12). □

Clearly, Theorem 4 follows as a corollary of Theorem 4.

A subtle point here is that of computational efficiency. Although designing polynomial time mechanisms is not our primary goal, it is clear that the only possible bottleneck is the last step of GENERAL-TWOQUERIES. Indeed, the mechanism runs in polynomial time whenever there is a polynomial-time algorithm (exact or $O(1)$-approximation) for the full information version of the corresponding optimization problem. The good news are that all variants of matching problems we presented can be solved efficiently by Edmond's algorithm (Edmonds, 1965) or its extensions (Marsh III, 1979).

**Corollary 5.** *There are deterministic polynomial-time mechanisms for General Graph Matching, Two-Sided Matching, General Graph k-Matching, and k-Constrained Resource Allocation which all use at most two queries per agent and have distortion $O(\sqrt{n})$.*

# B  Missing Material From Section 5

## B.1  Full Proof of Theorem 6

Let $\mathcal{M}$ be an arbitrary mechanism that makes at most $\lambda \geq 1$ queries per agent. Consider the following instance with $n$ agents and $m = n$ alternatives. We assume that $m$ satisfies the condition $m \geq \frac{1}{2} \sum_{\ell=1}^{\lambda} m^{(\lambda-\ell+1)/\lambda} + 2$, and also that it is superconstant; otherwise the theorem holds trivially. We partition the set of alternatives $\mathcal{A}$ into $\lambda + 2$ sets $A_1, A_2, \dots A_{\lambda+1}, A_{\lambda+2}$, such that

- $|A_\ell| = \frac{1}{2} m^{(\lambda-\ell+1)/\lambda}$ for $\ell \in [\lambda]$;
- $|A_{\lambda+1}| = 2$;
- $|A_{\lambda+2}| = m - \frac{1}{2} \sum_{\ell=1}^{\lambda} m^{(\lambda-\ell+1)/\lambda} - 2$.

The ordinal profile has the following properties:

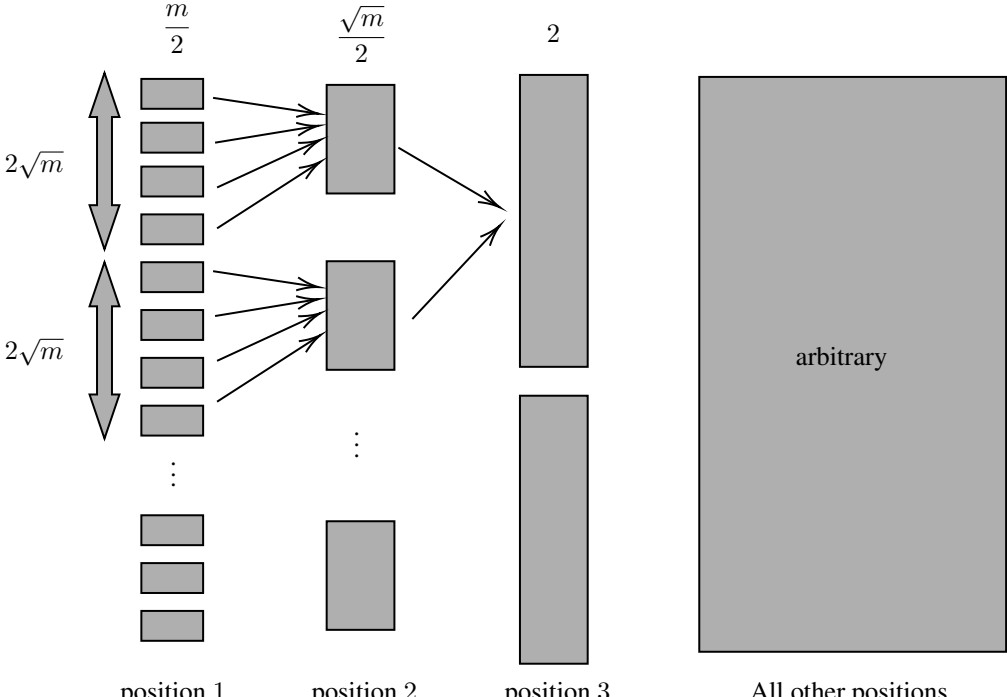

Figure 1: An overview of the instance used in the proof of Theorem 6 two queries ($\lambda = 2$). Each rectangle in the first three positions corresponds to an alternative. Each rectangle at position 1 contains two agents. Each rectangle at position 2 contains the agents from $2\sqrt{m}$ rectangles at position 1, as indicated by the arrows, meaning that those agents rank the same alternative second. The rectangles at position 3 contain $m/2$ agents each, corresponding to $\sqrt{m}/4$ rectangles at position 2. That is, the agents that rank second one of the first $\sqrt{m}/4$ alternatives at position 2, rank third the alternative corresponding to the first rectangle at position 3; similarly, the agents that rank second one of the last $\sqrt{m}/4$ alternatives at position 2, rank third the alternative corresponding to the second rectangle at position 3. The ranking of the alternatives in the remaining positions is consistent but otherwise arbitrary.

- For every $\ell \in [\lambda + 1]$, each alternative $j \in A_\ell$ is ranked at position $\ell$ by a set $T_{j,\ell}$ of $\frac{m}{|A_\ell|} = \Theta\left(m^{(\ell-1)/\lambda}\right)$ agents.

- For every $\ell \in [\lambda]$, every pair of agents that rank the same alternative in $A_\ell$ at position $\ell$, rank the same alternative in $A_{\ell+1}$ at position $\ell + 1$.

- For every agent, the alternatives that she does not rank in the first $\lambda + 1$ positions are ranked arbitrarily from position $\lambda + 2$ to $m$.

An example of the ordinal profile when $\lambda = 2$ is depicted in Figure 1 (see supplementary material). For every agent $i$, a query of $\mathcal{M}$ for alternative $j$ reveals a value of

- $m^{-\ell/\lambda}$ if $i$ ranks $j$ at position $\ell \in [\lambda + 1]$, and

- and a value of $0$ if $i$ ranks $j$ at any other position.

Given this instance as input, let $y$ be the alternative that $\mathcal{M}$ chooses as the winner. No matter the choice of $y$, we will define the cardinal profile so that it is consistent to the information revealed by the queries of $\mathcal{M}$, and the values of the agents for alternative $y$ are also consistent to the information that would have been revealed, irrespective of whether those values have actually been revealed. That is, any agent has a value of $m^{-\ell/\lambda}$ for $y$ if she ranks $y$ at position $\ell \in [\lambda + 1]$, and a value of $0$ if she ranks $y$ at any other position. Hence, the social welfare of $y$ is

- $\Theta\left(m^{(\ell-1)/\lambda}\right) \cdot m^{-\ell/\lambda} = \Theta(m^{-1/\lambda})$ if $y \in A_\ell$ for $\ell \in [\lambda + 1]$, or

- 0 if $y \in A_{\lambda+2}$.

Consequently, to show the desired bound of $\Omega(m^{1/\lambda})$ on the distortion of $\mathcal{M}$, it suffices to assume that $y \in A_\ell$ for some $\ell \in [\lambda + 1]$, and prove that the values of the agents that have not been revealed and do not correspond to alternative $y$ can always be defined such that there exists an alternative $x \neq y$ with social welfare $\Omega(1)$.

Suppose towards a contradiction that the cardinal profile cannot be defined in a way so that there exists an alternative $x$ with social welfare $\Omega(1)$. We make the following two observations:

(O1) If there exists an alternative $x \in A_1 \setminus \{y\}$ for which at least one agent in $T_{x,1}$ is *not* queried by $\mathcal{M}$ for $x$, then we can set the value of this agent for $x$ to be constant. Consequently, all the agents in $\bigcup_{j \in A_1 \setminus \{y\}} T_{j,1}$ that rank alternatives different that $y$ at position 1, must be queried at position 1.

(O2) Let $\varepsilon \in (0, 1)$ be a constant and $\ell \in \{2, \ldots, \lambda + 1\}$. Consider any alternative $x \in A_\ell \setminus \{y\}$ and any set of agents $S \subseteq T_{x,\ell}$ such that $|S| \geq \varepsilon \cdot \frac{m}{|A_\ell|} = \Theta(m^{(\ell-1)/\lambda})$. If at least $\frac{1}{2}|S|$ agents in $S$ are *not* queried by $\mathcal{M}$ for $x$, then we could set the value of all these agents for $x$ to be $m^{-(\ell-1)/\lambda}$ (which is the revealed value when $\mathcal{M}$ queries for alternatives ranked at position $\ell - 1$), and the social welfare of $x$ would be at least

$$\frac{1}{2}|S| \cdot m^{-(\ell-1)/\lambda} = \Theta(m^{(\ell-1)/\lambda}) \cdot m^{-(\ell-1)/\lambda} = \Theta(1).$$

Consequently, for every alternative $x \in A_\ell \setminus \{y\}$ and set $S \subseteq T_{x,\ell}$ such that $|S| \geq \varepsilon \cdot \frac{m}{|A_\ell|}$, at least $\frac{1}{2}|S|$ agents in $S$ must be queried at position $\ell$ for $x$.

Given these two observations, we are now ready to show by induction that the mechanism must make $\lambda + 1$ queries for a high proportion of the agents, contradicting that $\mathcal{M}$ makes at most $\lambda$ queries per agent.

For the base case, consider an alternative $x \in A_2 \setminus \{y\}$. By the definition of the ordinal profile, the agents in $T_{x,2}$ who rank $x$ at position 2 are partitioned into $\frac{|A_1|}{|A_2|}$ subsets such that all $\frac{m}{|A_1|}$ agents in each subset rank first the same alternative of $A_1$. By (O1) we have that, besides the agents that rank alternative $y$ at position 1, all other agents must be queried at position 1. Hence, there exists a set $S \subseteq T_{x,2}$ consisting of $|S| \geq \left(\frac{|A_1|}{|A_2|} - 1\right) \cdot \frac{m}{|A_1|}$ agents that are queried at position 1. By the definitions of $A_1$ and $A_2$, and since $m$ is superconstant, we have that $|S| \geq \frac{1}{2} \cdot \frac{m}{|A_2|}$. By (O2) for $\varepsilon = \frac{1}{2}$ and $\ell = 2$, we have that at least $\frac{1}{2}|S| \geq \frac{1}{4} \cdot \frac{m}{|A_2|}$ of the agents in $S$ must also be queried at position 2 for $x$.

Let $\ell \in \{3, \ldots, \lambda + 1\}$ and assume as induction hypothesis that for every alternative $z \in A_{\ell-1} \setminus \{y\}$ there is a set of agents $S_z \subseteq T_{z,\ell-1}$ such that $|S_z| \geq \frac{1}{2^{2(\ell-2)}} \cdot \frac{m}{|A_{\ell-1}|}$ who are queried by $\mathcal{M}$ at the first $\ell - 1$ positions. Consider an alternative $x \in A_\ell \setminus \{y\}$. By the definition of the ordinal profile, the agents in $T_{x,\ell}$ who rank alternative $x$ at position $\ell$ are partitioned into $\frac{|A_{\ell-1}|}{|A_\ell|}$ subsets such that all $\frac{m}{|A_{\ell-1}|}$ agents in each subset rank the same alternative in $A_{\ell-1}$ at position $\ell - 1$. So, by our induction hypothesis, there is a set $S \subseteq T_{x,\ell}$ consisting of

$$|S| \geq \left(\frac{|A_{\ell-1}|}{|A_\ell|} - 1\right) \cdot \frac{1}{2^{2(\ell-2)}} \cdot \frac{m}{|A_{\ell-1}|}$$

agents that are queried at the first $\ell - 1$ positions. By the definition of $A_{\ell-1}$ and $A_\ell$, and since $m$ is superconstant, we have that

$$|S| \geq \frac{1}{2^{2(\ell-2)+1}} \cdot \frac{m}{|A_\ell|}.$$

Since $\ell \leq \lambda + 1$ and $\lambda$ is a constant, by observation (O2) for $\varepsilon = \frac{1}{2^{2(\ell-2)+1}}$, we have that at least

$$\frac{1}{2}|S| \geq \frac{1}{2^{2(\ell-1)}} \cdot \frac{m}{|A_\ell|}$$

agents in $S$ must also be queried at position $\ell$ for $x$.

Now, let $x \in A_{k+1} \setminus \{y\}$. The above induction shows that there are at least $\frac{1}{2^{2\lambda}} \cdot \frac{m}{|A_{\lambda+1}|}$ agents in $T_{x,\lambda+1}$ who must be queried by $\mathcal{M}$ at the first $\lambda + 1$ positions. This contradicts the fact that $\mathcal{M}$ can make at most $k$ queries per agent, and the theorem follows. □

## B.2   Proof of Theorem 7

Consider any social choice instance with valuation profile $\mathbf{v}$ that induces the ordinal preference profile $\succ_\mathbf{v}$. Let $y$ be the alternative chosen by the mechanism when given as input this instance, and denote by $x$ the optimal alternative. We will show that $\mathrm{SW}(x) \leq (1 + (1 + c) \cdot \sqrt{m})\mathrm{SW}_R(y)$. The bound on the distortion will then follow by the obvious fact that $\mathrm{SW}(y) \geq \mathrm{SW}_R(y)$.

We can write the optimal welfare as

$$
\begin{aligned}
\mathrm{SW}(x) &= \mathrm{SW}_R(x) + \mathrm{SW}_C(x) \\
&\leq \mathrm{SW}_R(y) + \mathrm{SW}_C(x),
\end{aligned} \tag{13}
$$

where $\mathrm{SW}_C(x)$ is the concealed welfare of $x$, consisting of the values of agents for $x$ that were not revealed by the queries of the mechanism, and the inequality follows by the fact that $y$ is the alternative that maximizes the revealed welfare. Let $S$ be the set of agents who were *not* queried about their value for $x$, and partition $S$ into the following two subsets:

- $S^{\geq}$ consists of the agents in $S$ for whom the second query is about an alternative that the agent considers *better* than $x$;

- $S^{<}$ consists of the agents in $S$ for whom the second query is about an alternative that the agent considers *worse* than $x$.

Given these sets, now let

$$
\mathrm{SW}_{\overline{C}}^{\geq}(x) = \sum_{i \in S^{\geq}} v_{i,x} \quad \text{and} \quad \mathrm{SW}_{\overline{C}}^{<}(x) = \sum_{i \in S^{<}} v_{i,x}.
$$

be the contribution of the agents in $S^{\geq}$ and of the agents in $S^{<}$ to the concealed welfare of $x$, respectively. That is,

$$
\mathrm{SW}_C(x) = \mathrm{SW}_{\overline{C}}^{\geq}(x) + \mathrm{SW}_{\overline{C}}^{<}(x).
$$

By the definition of the mechanism, each agent is queried about her favorite alternative in the sufficiently representative set $B$. For every $j \in B \setminus \{x\}$, let $S_j^{\geq} \subseteq S^{\geq}$ be the set of agents in $S^{\geq}$ who are queried for alternative $j$ instead of $x$. Thus, $S^{\geq} = \bigcup_{j \in B \setminus \{x\}} S_j^{\geq}$. By the definition of $S^{\geq}$, the fact that $y$ maximizes the revealed welfare, and since $|B| \leq c \cdot \sqrt{m}$, we obtain

$$
\begin{aligned}
\mathrm{SW}_{\overline{C}}^{\geq}(x) &= \sum_{j \in B \setminus \{x\}} \sum_{i \in S_j^{\geq}} v_{i,x} \\
&\leq \sum_{j \in B \setminus \{x\}} \sum_{i \in S_j^{\geq}} v_{i,j} \\
&\leq \sum_{j \in B \setminus \{x\}} \mathrm{SW}_R(j|\mathbf{v}) \\
&\leq |B| \cdot \mathrm{SW}_R(y|\mathbf{v}) \\
&\leq c \cdot \sqrt{m} \cdot \mathrm{SW}_R(y).
\end{aligned} \tag{14}
$$

Since all the agents in $S^{<}$ are queried for alternatives in the sufficiently representative set $B$ that they consider worse than $x$ and $B$, it must be the case that $|S^{<}| \leq \sqrt{m}$. Since all agents are queried at the

first position for their favorite alternative, we obtain

$$
\begin{aligned}
\mathrm{SW}_C^{\leq}(x) &= \sum_{i \in S^<} v_{i,x} \\
&\leq \sum_{i \in S^<} \max_{j \in \mathcal{A}} v_{i,j} \\
&\leq |S^<| \cdot \max_{i \in S^<} \max_{j \in \mathcal{A}} v_{i,j} \\
&\leq \sqrt{m} \cdot \mathrm{SW}_R(y).
\end{aligned}
\tag{15}
$$

The bound now follows by (13), (14) and (15). $\qquad\square$