# OpenReview forum: "Don't Roll the Dice, Ask Twice: The Two-Query Distortion of Matching Problems and Beyond"
_NeurIPS.cc/2022/Conference — NeurIPS 2022 Accept_

### Official Review · Reviewer_NGwA · 2022-06-30

**Rating:** 7
**Confidence:** 3
**Soundness:** 4 excellent
**Presentation:** 4 excellent
**Contribution:** 3 good

**Summary:**

This paper studies social welfare maximization when given only ordinal information for each agent, but where we are allowed to make *two* cardinal queries per agent.  They focus on "one-sided matching", but their results also extend more generally.  In slightly more detail, suppose that every agent has a value for every item, and our goal is to find the matching maximizing the social welfare.  However, we are not given the value ("cardinal") information, but are instead only given the *ordering* for every agent.  We can then design mechanisms that use only this ordinal information, and use as our measure of quality the worst case over all valuations of the optimal social welfare (matching) vs the welfare achieved by the mechanism.  This is called the "distortion".  This has been studied extensively over the past decade, and much is known.

This paper continues a new line of work where we augment our mechanisms with the ability to make a small number of valuation ("cardinal") queries.  A previous paper showed that with $O(\log n)$ queries per agent we can get $O(^{1/k})$ distortion for any constant k that we want, and it was also shown that with k queries the best distortion we can hope for is $\Omega(n^{1/k})$.  So there was an obvious open question: can we get sublinear distortion with less than $\log n$ queries?  In the extreme: what distortion is achievable with only *two* queries?

This paper resolves the two-query case, showing that $O(n^{1/2})$ distortion is achievable.  They do this with a deterministic algorithm and with no assumptions on the valuations.  Previous sublinear distortion bounds (e.g., without any queries) required either randomness, assumptions on the valuations, or both.  Their algorithm first queries the valuation of every agent's highest-rank item.  They then use the ordinal information to compute a "sufficiently representative assignment" A -- this is basically a many-one assignment where every item is matched to at most $n^{1/2}$ agents, and also there are at most $n^{1/2}$ agents that prefer what they are matched with in OPT to what they are matched to in the assignment.  The authors show that a simple algorithm can be used to generate such an assignment using only the ordinal information.  Then for the second query, they query the valuation for every agent of the item they're matched to in A.  Finally, they compute an optimal solution that uses only the valuations from these two queries (all non-queried valuations are set to 0).

In the analysis, they show that the dual $n^{1/2}$ guarantees of the sufficiently representative assignment can be used to balance the cost both of the agents that query a "too high" item (an item that they prefer to what they get in OPT) and of the agents that query a "too low" item.

They then study more general settings than just one-sided matching.  First, they study a wider variety of graph problems which they call "matching extended k-families", and which include pretty natural problems such as two-sided matching and general graph k-matchings.  Technically, this type of problem does not include the one-sided matching case, since they assume in their graphs that every vertex is an agent.  So the cannot just re-use the exact same ideas and results.  However, they show that an $\Omega(n^{1/k})$ lower bound still holds, and that a similar algorithm with a more complicated argument also gives $O(n^{1/2})$ distortion with only two queries per agent.

Finally, they study the most general social choice setting, where there are $n$ agents, $m$ alternatives, every agent has a valuation for every alternative, and we want to choose the alternative that maximizes social welfare.  Here they show an $\Omega(m^{1/k})$ lower bound on deterministic mechanisms that make at most $k$ queries, and give an algorithm which is essentially a generalization of their previous algorithms which gives distortion $O(\sqrt{m})$ as long as $m = \Omega(n)$.


**Questions:**

- You list handling a constant number $>2$ queries as an open problem.  It would be nice to discuss this a bit more, and in particular discuss why your approach can't already achieve this.

- You also mention that for the general social choice problem, you suspect that adaptivity is necessary.  It would be nice to expand on this.  Why do you suspect that adaptivity is necessary?  Could adaptivity be used even in the grpah case to get better results with 3 or more queries?


**Limitations:**

This is a theory paper, so I think the discussion of limitations and negative social impact is sufficient.


**Strengths And Weaknesses:**

Strengths:
- This is a pretty natural set of questions.  Distortion of having only ordinal information has been well-studied and seems to be pretty well-motivated.  But since we have strong lower bounds in that setting, it's natural to try to go "beyond worst-case" by allowing extra information.  A small number of value queries seems to me to be a pretty natural way to do this (of course it's not the only possible way of going beyond worst-case, but I think it's an interesting one).  And this is not the first paper which studies this setting.

- The main upper bound seems quite strong to me.  The problem is particularly natural (one-sided matching), it matches a known lower bound, the algorithm is quite simple but requires some non-obvious analysis, and they show that two queries are already enough to give nontrivial distortion guarantees.  It feels to me like the "right" result, for a natural and interesting problem.

- The generalizations are also quite interesting.  The problems are very natural (particularly two-sided matching and general social choice), and the results are pretty strong (particularly for two-sided matching and the other graph-based generalizations, where no extra assumptions are needed).  And the algorithms are basically the same as the one-sided matching case, but with slight tweaks and more complicated analyses.  So they're generalizations in a sense that I like a lot -- they reinforce the main ideas of the paper, rather than giving an entirely new set of ideas, but do add significant technical difficulty and complexity.

Weaknesses:
- The main result is quite strong (as I discussed above), but is actually a little weaker than I was hoping for.  This is because the result only holds for two-queries.  In many algorithmic settings, there is a tradeoff between some parameter and some notion of "quality", and when there's not such a tradeoff there is some kind of "phase transition".  Based on the known lower bound, I was hoping that this paper have an $O(n^{1/k})$-distortion upper bound for $k$ queries (giving a tradeoff), or alternatively showed that it was not possible to get below $n^{1/2}$ queries without using superconstant queries (showing a phase transition).  But this paper didn't give any results on constant queries beyond 2, which I found a little disappointing.  However, it's worth pointing out that two is a particularly interesting case, since (as the authors point out) it is the smallest number of queries for which sublinear bounds are possible.  I just wish the authors had either given more general results, or discussed why their approach doesn't / can't give such results.

Overall, I really liked this paper, and think it should be accepted.

---

> ### Author Response · Authors · 2022-07-31
> **Responses to questions**
>
> -- You list handling a constant number > 2 queries as an open problem. It would be nice to discuss this a bit more, and in particular discuss why your approach can't already achieve this.
>
> Re: To obtain the bound of $O(\sqrt{n})$ with two queries, our approach balances out two quantities: the number of agents that are assigned (in the sufficient representative assignment A; see Def. 2) to something less preferred than what they receive in X, and the number of agents assigned to each item in A. When both of these quantities are $O(\sqrt{n})$, then this can be done.
>
> To extend this technique to 3 queries to achieve a distortion of $O(n^{1/3})$, we would have to use the second query to “split” the set of agents (which is of course known) into two uneven sets of size $n^{2/3}$ and $n^{1/3}$ respectively, as before. Then, in the next step, using the third query, we could try to once again apply the same approach as above to further “split” the set of size $n^{2/3}$ into two even subsets of size $n^{2/3}$ agents. However, at this point we do not know who these $n^{2/3}$ agents would be, and therefore this is not possible.
>
>
> -- You also mention that for the general social choice problem, you suspect that adaptivity is necessary. It would be nice to expand on this. Why do you suspect that adaptivity is necessary? Could adaptivity be used even in the graph case to get better results with 3 or more queries?
>
> Re: The social choice setting is more challenging than the matching setting, because it is less structured. In fact, our techniques are enough to show a bound of $O(\sqrt{n})$ for social choice, which translates to a bound of $O(\sqrt{m})$ in some cases only. The relation between $m$ and $n$ seems to play an intricate role here, which is why we think that our technique has reached its limits.
>
> By “adaptivity”, we mean that the algorithm will decide which queries to ask based on the ordinal information as well as the history of answers to previous queries (i.e., decide how to use the third query based on the ranking and the values learnt from the first two queries). Our algorithms only use the ordinal information to decide where to make queries, which seems, to us, insufficient for achieving the best possible bound in the social choice setting.
>
> Similarly, the reviewer indeed has the same intuition as we do: for 3 or more queries in the matching setting, again we would need to use some adaptive queries, as our techniques seem to fall short as we explain in the answer to the first question above.

---

### Official Review · Reviewer_WLMQ · 2022-07-10

**Rating:** 4
**Confidence:** 1
**Soundness:** 4 excellent
**Presentation:** 4 excellent
**Contribution:** 2 fair

**Summary:**

The paper deals with optimizing social welfare given preference orderings of agents. The paper improves social welfare by asking each agent two additional questions and it shows that social welfare can be optimized compared to knowing all agents' exact valuations up to a divisor of SQRT(n).
Based on the solution to this problem further statements are derived.

**Questions:**

none

**Limitations:**

While the paper offers clear progress wrt underlying theory,
the paper could have discussed practical applications and applicability at least briefly.



**Strengths And Weaknesses:**

Strengths:
+ Novel algorithm and theorems about social welfare optimzation given additional queries
+ Useful extensions beyond the core approach
Weakness:
+ The main comparisons are constituted by logarithmically many queries. For most practical purposes, logarithm is like a constant. A practically relevant comparison should therefore show the advantage also empirically by experiments.
+ I don't see this as a relevant paper for a machine learning conference

---

> ### Author Response · Authors · 2022-07-31
> **Response to comments**
>
> -- Relevance to NeurIPS
>
> Re: We would like to point out that NeurIPS regularly accepts works that are related to the theoretical foundations of ML and more broadly of AI. There is a significant number of matching / computational social choice papers which are not directly related to learning. From NeurIPS 2021 alone, a representative sample is:
>
> - Lirong Xia: The Semi-Random Satisfaction of Voting Axioms
>
> - Nathan Noiry, Vianney Perchet, Flore Sentenac: Online Matching in Sparse Random Graphs: Non-Asymptotic Performances of Greedy Algorithm
>
> - Joshua Kavner, Lirong Xia: Strategic Behavior is Bliss: Iterative Voting Improves Social Welfare
>
> - Brian Brubach, Nathaniel Grammel, Will Ma, Aravind Srinivasan: Improved Guarantees for Offline Stochastic Matching via new Ordered Contention Resolution Schemes
>
> - Grant Schoenebeck, Biaoshuai Tao: Wisdom of the Crowd Voting: Truthful Aggregation of Voter Information and Preferences
>
> On the topic of distortion in particular, reference [28] in our submission is a NeurIPS paper as well.
>
> -- Experiments
>
> Re: While we conceptually disagree with the statement that the logarithm is essentially a constant in practice in a setting where asking cardinal queries can be cognitively demanding, we do agree that having some experimental results could be a nice complement to our main theoretical results. We would like to point out, however, that such experimental results are scarce in the distortion literature in general. We do not think that simply running experiments on randomly generated instances (e.g., drawn from simple distributions) and adding them to our paper would necessarily be of much practical relevance for the real world. On the other hand, data about cardinal preferences are very limited and often not publicly available. An experimental approach would certainly be very useful, but it seems like the topic of a separate paper, which will build upon the theoretical results of the literature (including the ones we provide here).
>
> That being said, we are not entirely sure what kind of experiments the reviewer has in mind for comparing the two vs logarithmically-many queries algorithms. The advantage of our algorithm by default lies in the number of queries (2 vs log-many), not in the distortion (as both algorithms achieve asymptotically the same bound).  Perhaps it could give us some intuition about the constants hidden in the O(sqrt n) bounds, for some families of instances used in the experiments.
>
>
> -- On applications:
>
> Re: Over the years, matching problems have found numerous important applications in practice, such as residents matching [1], college admissions and kidney exchange (e.g., see [2]). Importantly, the algorithms employed for these applications are purely ordinal that have been proposed in the associated social choice literature. See also the discussion paper [3] on the application of (ordinal) matching algorithms in school choice in Amsterdam since 2005.
>
> [1] https://en.wikipedia.org/wiki/National_Resident_Matching_Program
>
> [2] https://qz.com/421547/nobel-prize-winner-alvin-roth-explains-the-hidden-economics-behind-tinder-marriage-and-college-admissions/
>
> [3] https://docs.iza.org/dp9118.pdf

---

### Official Review · Reviewer_B6to · 2022-07-11

**Rating:** 4
**Confidence:** 4
**Soundness:** 4 excellent
**Presentation:** 3 good
**Contribution:** 2 fair

**Summary:**

The authors propose a framework that approximates a matching problem using only two queries per agent.
The authors show that they can achieve O(sqrt(n)) approximation guarantee, and show that this is the best
they can do given the queries.

**Questions:**

The two queries algorithm is allowed to ask are very different. The first asks the most preferred item,
while the second asks the utility of a certain item. How this compares to the existing work? How this extends
to lambda > 2 queries.

**Limitations:**

The available information is so limited, that the algorithm cannot do much here. Essentially, the algorithm
will give the preferred option or a random choice. This hints that in practice two queries are too little
to produce any interesting result. Perhaps developing an algoriithm for higher lambdas would be more fruitful.


**Strengths And Weaknesses:**

+ Theoretically interesting result. Especially the tightness result completes the picture.
- The amount of information is so limited that the algorithm cannot  do too much.
- Limited applications

---

> ### Author Response · Authors · 2022-07-31
> **Response to questions and comments**
>
> – “The available information is so limited, that the algorithm cannot do much here”.
>
> Re: In the distortion literature, the algorithms typically only have access to ordinal information, and such algorithms are also being used in practice for several applications. Our algorithms use *more* information than those (ordinal information + 2 queries), which is why they can do more in terms of the distortion. We do not agree that our algorithm does not “do much”. It utilizes an elegant combinatorial idea and manages to achieve distortion $O(\sqrt{n})$, without using any randomization or normalization. In contrast, in the standard ordinal setting (without queries), the best possible algorithms, using both randomization and normalization, still cannot do better than $O(\sqrt{n})$. In that sense, our $O(\sqrt{n})$ bound is very meaningful in the context of this literature.
>
> – “Essentially, the algorithm will give the preferred option or a random choice.”
>
> Re: Please note that in the final matching computed by our algorithm, most agents will *not* be assigned their favorite choice in most preference profiles (for example, consider instances where most agents agree on which item is the best – only one of them could get it). Also, there is no randomness in our setting, we focus only on deterministic algorithms, which is clearly stated in the Introduction (for instance, see line 75 on page 2). Thus, it is not clear what the Reviewer means by “a random choice”. Anything along the lines of what the Reviewer is suggesting (try to assign to each agent her best item or give something arbitrarily) will only achieve a $\Theta(n)$ distortion with two queries.
>
> – How this compares to the existing work? How this extends to lambda > 2 queries.
>
> Re: As we mention in lines 157-158 “without any normalization assumptions [...] a mechanism cannot have any guarantee unless it queries every agent about her favorite item”. Thus, *any* mechanism asking 2 or more queries and having bounded distortion must spend one query on the first position of each agent’s preference ranking. This is true for existing mechanisms as well. As we discuss in our Introduction, previous work showed that it is possible to achieve distortion $O(\sqrt{n})$ with $O(\log{n})$ queries per agent, while for 2 queries specifically, the best known algorithm before our work achieves a distortion of $O(n^{2/3} \cdot \sqrt{\log{n}})$ for unit-sum valuations. Without normalization (like in our setting), the best known bound for 2 (or any constant number of) queries was $\Theta(n)$. In all cases, even in the result assuming normalization, one query per agent is used on the top choice of each agent. The technically challenging part is choosing how to ask the remaining queries. In that respect, our approach provides not only a very significant improvement over the previous work, but also a novel perspective on how to make such choices.
>
> Extending our algorithm for lambda > 2 is a challenging open question, and seems to require new techniques. Please see our response to a relevant question of Reviewer NGwA for some intuition about the difficulty of extending our techniques for more queries.
>
> – Limited applications.
>
> Re: Over the years, matching problems have found numerous important applications in practice, such as residents matching [1], college admissions and kidney exchange (e.g., see [2]). Importantly, the algorithms employed for these applications are purely ordinal that have been proposed in the associated social choice literature. See also the discussion paper [3] on the application of (ordinal) matching algorithms in school choice in Amsterdam since 2005.
>
> [1] https://en.wikipedia.org/wiki/National_Resident_Matching_Program
>
> [2] https://qz.com/421547/nobel-prize-winner-alvin-roth-explains-the-hidden-economics-behind-tinder-marriage-and-college-admissions/
>
> [3] https://docs.iza.org/dp9118.pdf

---

### Official Review · Reviewer_p557 · 2022-07-11

**Rating:** 7
**Confidence:** 2
**Soundness:** 3 good
**Presentation:** 4 excellent
**Contribution:** 4 excellent

**Summary:**

The manuscript tackles the problem of maximising social welfare amongst agents for assignments of items to agents when agents only disclose their preference ordering of items (base on a hidden score of each item). The manuscript focuses strongly on the distortion of a matching mechanism, i.e., the worst-case ratio between achieved and optimal social welfare amongst all possible valuation profiles. One major result is that asking agents first for their favourite alternative, then computing a representative subset and asking for their favourite alternative amongst the subset yields a $O(\sqrt{n})$ distortion ($n$ is both the number of items and agents). Previously, such a result required randomization and normalization of agent values, whereas the novel result is deterministic and obiviates the need for normalization and thereby extends the applicability of the result. This comes only at the cost of having to elicit preference twice from agents, which seems acceptable in most settings. The challenging part of developing the approach lies in the computation of the representative subset to allow for the resulting analysis.


**Questions:**

Q1. What are some examples of practical applications mentioned in Section C of the supplementary material?

**Ethics Review Area:**

["Discrimination / Bias / Fairness Concerns"]

**Limitations:**

Specifically, as Section C in the supplementary highlights performance in practice, it would be useful to add a small (simple) empirical study with simulated agents with some prior randomised approach as a baseline and mention a few concrete examples of potential applications.

**Strengths And Weaknesses:**

STRENGTHS

S1. Motivation/Relevance: Interesting, very general problem of how to match agents with items based only on ordinal preference indications (which is generally easier to elicit).

S2. Novelty/Significance: The basic idea of eliciting preferences twice, first on all items and then a representative subset seems novel and a simple, elegant solution

S3. Presentation/Soundness/Related Work: The work is well-written, seems to substantiate all claims (very extensive supplementary material) and cover the relevant related work.

S4. The work discusses a wider range of variants and both possibility results as well as achieved results.

WEAKNESSES

W1. An empirical study could make the paper more relevant to more practically oriented readers.

W2. Although it would seem to be general enough to cover a wider range of applications, some examples for practical applications could be beneficial

---

> ### Author Response · Authors · 2022-07-31
> **Responses to questions and comments**
>
> Re Q1 and W2: Over the years, matching problems have found numerous important applications in practice, such as residents matching [1], college admissions and kidney exchange (e.g., see [2]). Importantly, the algorithms employed for these applications are purely ordinal that have been proposed in the associated social choice literature. See also the discussion paper [3] on the application of (ordinal) matching algorithms in school choice in Amsterdam since 2005.
>
> [1] https://en.wikipedia.org/wiki/National_Resident_Matching_Program
>
> [2] https://qz.com/421547/nobel-prize-winner-alvin-roth-explains-the-hidden-economics-behind-tinder-marriage-and-college-admissions/
>
> [3] https://docs.iza.org/dp9118.pdf
>
> Re W1: We agree that having some experimental results could be a nice complement to our main theoretical results. We would like to point out, however, that such experimental results are scarce in the distortion literature in general. We do not think that simply running experiments on randomly generated instances (e.g., drawn from simple distributions) and adding them to our paper would necessarily be of much practical relevance for the real world. What we need is a systematic approach that perhaps uses some real-world data and performs extensive experiments which start with the original distortion settings (without queries) and then consider the settings with queries as well. This is challenging, because data about cardinal preferences are very limited and often not publicly available. So while an experimental approach could certainly be very useful, it seems like the topic of a separate paper, which will build upon the theoretical results of the literature (including the ones we provide here).

---

> > ### Comment · Reviewer_p557 · 2022-08-05
> > **Strategic voting**
> >
> > Thank you for the response. With regards to NRMP [1], how do the proposed methods fare in terms of susceptibility to "strategic voting" (misrepresenting preferences)?

---

> > > ### Author Response · Authors · 2022-08-05
> > > **Response to Strategic Voting**
> > >
> > > The particular mechanisms we have designed in this paper are not strategyproof. Strategyproofness, as well as equilibrium efficiency (price of anarchy), has been considered in the context of distortion in previous works for matching (see reference [20] in the paper) with strategic agents. There, it was shown that the best possible distortion achievable by strategyproof mechanisms and the best possible price of anarchy of any mechanism is achieved by ordinal mechanisms. In other words, even if we had access to the full cardinal (numerical) information about the preferences, strategic behavior imposes constraints that only allow us to use the ordinal information. Essentially, improved distortion bounds via queries and incentive robustness (strategyproofness) are incompatible.

---

> > > > ### Comment · Reviewer_p557 · 2022-08-08
> > > > **Strategyproof**
> > > >
> > > > Thank you for all the responses, they have been quite insightful.

---

### Meta-Review · Area_Chair_XzrT · 2022-08-29

**Recommendation:** Accept
**Confidence:** Certain

**Metareview:**

This work studies a narrow, but important problem of how much cardinal information is needed to achieve near optimal matchings. The authors show that with just two queries (one is required for any non-trivial results) they can achieve non-trivial results in a very general setting. Moreover, they show that their results are tight.

**Award:**

No

---

### Decision · Program_Chairs · 2022-09-14

Accept